# NoPo-Avatar: Generalizable and Animatable Avatars from Sparse Inputs without Human Poses

**Jing Wen**   **Alexander G. Schwing**   **Shenlong Wang**
University of Illinois Urbana-Champaign
{jw116, aschwing, shenlong}@illinois.edu
https://wenj.github.io/NoPo-Avatar/

## Abstract

We tackle the task of recovering an animatable 3D human avatar from a single or a sparse set of images. For this task, beyond a set of images, many prior state-of-the-art methods use accurate "ground-truth" camera poses and human poses as input to guide reconstruction at test-time. We show that pose-dependent reconstruction degrades results significantly if pose estimates are noisy. To overcome this, we introduce NoPo-Avatar, which reconstructs avatars solely from images, without any pose input. By removing the dependence of test-time reconstruction on human poses, NoPo-Avatar is not affected by noisy human pose estimates, making it more widely applicable. Experiments on challenging THuman2.0, XHuman, and HuGe100K data show that NoPo-Avatar outperforms existing baselines in practical settings (without ground-truth poses) and delivers comparable results in lab settings (with ground-truth poses).

## 1   Introduction

Animatable human rendering aims to 1) reconstruct an animatable 3D representation from given images, and to 2) synthesize novel views of possible novel human poses. The task is of great utility in VR/AR applications. Recent advances in rendering techniques have significantly improved the realism and fidelity of the rendered results.

Concretely, recent generalizable human rendering approaches [8–10, 28] reconstruct an animatable representation in a single deep net feed-forward pass. This significantly speeds up the reconstruction to subsecond levels without compromising the rendering quality. In addition, large-scale training enables these methods to perform well even with very sparse inputs. These advances make generalizable human rendering more practical for real-world applications compared to per-scene optimized methods. Despite the pros, most generalizable human rendering approaches assume that accurate camera poses and human poses are available for reconstruction at test-time. These input poses serve as strong guidance in locating the correspondences and gathering the aligned image features. However, the use of accurate input poses during test-time reconstruction introduces a challenge, as illustrated in Fig. 1: we assess the dependence of the rendering quality on test-time input pose quality used for reconstruction, either by 1) injecting Gaussian noise of different standard deviations into accurate ground-truth test-time input poses used for reconstruction; or by 2) using a predicted pose for reconstruction at test-time. For the results shown in Fig. 1, we always used ground-truth poses for test-time rendering. The noisier the test-time poses used for reconstruction, the worse the rendering quality the methods achieve. For in-the-wild scenarios where poses are estimated, the performance of existing methods degrades significantly, as shown in Fig. 1(a) horizontal axis label "pred". This sensitivity to input poses is not a desirable property. Ideally, we expect the method to produce consistent and high-quality results, no matter the quality of the input poses at reconstruction.

39th Conference on Neural Information Processing Systems (NeurIPS 2025).

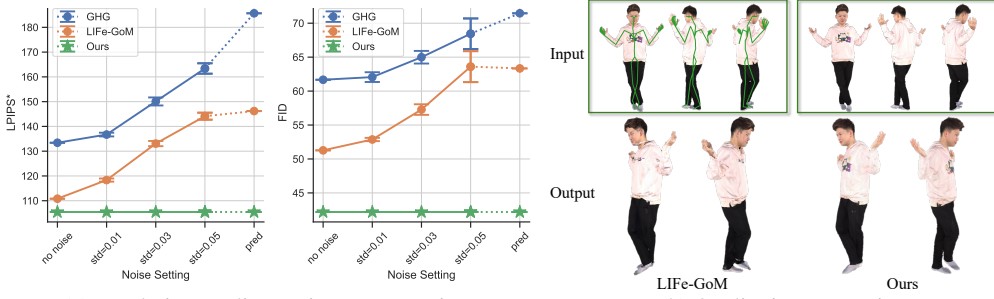

(a) Rendering quality vs. input-pose noises

(b) Qualitative comparisons

Figure 1: (a) **Sensitivity to input pose noises.** Previous methods [10, 28] take camera poses and human poses as inputs. We measure their sensitivity to input poses by injecting Gaussian noise of different standard deviations or using a predicted pose. (Averaged over 5 runs for Gaussian noise; std are multiplied by 3 for better visualization.) (b) **Comparisons on rendering quality.** With the predicted inaccurate input poses, LIFe-GoM cannot produce high-fidelity rendering. In contrast, our methods, which does not take any poses as inputs, produce high-quality rendering.

To achieve this, we develop NoPo-Avatar, which completely eliminates the dependence of the animatable avatar reconstruction on camera poses and human poses. Hence, our reconstruction and the subsequent rendering is not affected by the quality of the input poses used for reconstruction, as shown in Fig. 1(a). Importantly, we represent our reconstruction in the canonical T-pose, which can be animated to arbitrary novel poses without any post-processing. To obtain the canonical T-pose representation, we design a dual-branch model that captures observed details and inpaints missing regions. Specifically, our model consists of two types of branches, a template branch and image branches. The template branch starts from an encoding of the shape in the canonical pose and outputs Gaussians relative to the average SMPL-X template in T-pose. The image branches predict pixel-aligned Gaussians (splatter images) in the same coordinate system as the template branch. Not only can we model the fine details in the observed input images via the spatter images obtained from the image branches, but we can also inpaint unseen regions via the template branch. For this, we adopt an encoder-decoder architecture, similar to NoPoSplat [34]: the template and the input images are independently embedded in each encoder and interact with each other via cross-attention in the decoder. Our model is trained end-to-end using photometric losses and auxiliary regularization on Gaussians' 3D positions as well as linear blend skinning weights. We note that recent works such as LHM [16] and IDOL [37] scale training to improve generalization, and, like our method, eliminate the need for pose data. However, unlike IDOL and LHM which operate on a single input image, we support multiple input views. More importantly, we introduce a dual-branch design. In contrast, LHM and IDOL rely solely on a template branch, which subtly differs from ours in the template encoding and the fusion of template embeddings and image features. We find the image branch enhances reconstruction of details in observed regions.

We evaluate our method on THuman2.0, XHuman and HuGe100K and compare to the state-of-the-arts. Our method reconstructs the avatars in high quality without any pose priors, significantly outperforming state-of-the-arts that use predicted poses, as shown in Fig. 1(b), and on par with those that use ground-truth poses.

Our contributions are twofold:

- We propose NoPo-Avatar, a novel model that reconstructs an animatable human avatar given only input images. Our model does not use camera and human poses.
- We demonstrate that the reconstruction obtained without camera and human poses leads to high-quality novel view and novel pose rendering.

## 2   Related Work

**Generalizable human rendering from sparse inputs or a single image.** Generalizable methods learn inductive biases from large-scale datasets, enabling use of these methods in case of sparse inputs, or even a single image. We categorize methods tackling this task into one-stage and multi-stage methods. Multi-stage methods [29, 7, 2, 12, 5, 31, 25] usually generate multiview images or videos using a 2D generative model such as a diffusion model, and then reconstruct the 3D representation

from the multiview images. One-stage methods adopt a feed-forward neural network to produce the 3D representation [8, 9, 36, 10, 28, 38, 33, 4, 37, 16] or directly generate the representation with a diffusion model given a single image as the condition [32]. Among these works, several [4, 9, 28, 16, 5, 38] reconstruct in a canonical pose space, which enables animatation without post-processing such as skeleton binding. Our approach is a one-stage method and requires only a single feed-forward pass for reconstruction. We also reconstruct in the canonical pose space to support animation.

To maximize the use of human priors, most of the recent work [4, 9, 10, 28, 38, 33] takes the posed SMPL [11] or SMPL-X [14] meshes as input and assumes that accurate human poses and camera poses are available during test-time reconstruction. The SMPL/SMPL-X poses are used to sample aligned features from the inputs explicitly or implicitly. While this assumption eases the task, it also limits generalization to in-the-wild subjects, as accurate pose estimation either takes too long or estimated poses are not accurate enough. More recently, RoGSplat [30] tackles the task of generalizable rendering with inaccurate pose estimation. Differently, we eliminate the need for both human pose and camera pose altogether and only operate on images and subject masks. Consequently, our method is not only insensitive to the accuracy of input poses, but also saves the time required for pose estimation.. The very recent two works IDOL [37] and LHM [16] also consider reconstructing in the canonical pose space without use of human poses and camera poses. Our method differs from IDOL and LHM in two main ways: 1) Our method operates on any number of input images, while IDOL and LHM only address the single image setting, leaving an extension to multiple images open. 2) We predict two sets of Gaussians, one in the projected UV space and the other aligned with all foreground pixels in the input images, while IDOL and LHM only predict Gaussians in SMPL/SMPL-X's projected UV space. This enables our method to model the observed details.

**Generalizable scene rendering without pose priors.** Our setup aligns with recent efforts in generalizable scene rendering without pose priors. Note that pose priors here refer to camera poses only. The success of DUSt3R [26] sheds light on pixel-aligned geometry estimation without knowledge of the relative camera poses. This idea was adopted by generaliable rendering [34, 3, 24], since inaccurate camera localization can potentially lead to corrupted renderings.

While the above works reconstruct the scene in one input image's coordinate system, our task is inherently more challenging: the reconstruction is in a canonical T-pose which differs from the human poses in input images. Use of the T-pose reconstruction is important as it enables us to animate new poses without post processing.

## 3  NoPo-Avatar

Given input images $\{\mathbf{I}_n\}_{n=1}^N, \mathbf{I}_n \in \mathbb{R}^{H^I \times W^I \times 3}$, and masks $\{\mathbf{M}_n\}_{n=1}^N, \mathbf{M}_n \in \{0,1\}^{H^I \times W^I}$, indicating the pixels which correspond to a human subject, we aim to synthesize novel views/poses given target camera extrinsics $\mathbf{E}$, intrinsic $\mathbf{K}$, and the target human shape and pose $\mathbf{P} = (\boldsymbol{\beta}, \boldsymbol{\theta})$. Here, $N$ is the number of input images, and $H^I$ and $W^I$ denote the height and width of the input images. The human shape $\boldsymbol{\beta}$ and pose $\boldsymbol{\theta}$ are represented in SMPL-X's format. Importantly, unlike prior work [36, 10, 28], our method does not use any human poses and camera poses of the input images.

To achieve this goal, we develop NoPo-Avatar, which consists of two modules: reconstruction and rendering. Given target camera intrinsics $\mathbf{K}$, extrinsics $\mathbf{E}$, and human poses $\mathbf{P}$, the rendering module (Render) computes the target image and alpha opacity mask:

$$(\mathbf{I}, \mathbf{M}) = \texttt{Render}(\mathcal{G}; \mathbf{E}, \mathbf{K}, \mathbf{P}) \tag{1}$$

from the reconstructed canonical 3D T-pose representation $\mathcal{G}$. It is the output of the reconstruction module (Recon), which operates on input images and subject masks, i.e.,

$$\mathcal{G} = \mathcal{G}^T \cup \mathcal{G}^I = \texttt{Recon}(\{\mathbf{I}_n\}_{n=1}^N, \{\mathbf{M}_n\}_{n=1}^N). \tag{2}$$

The reconstructed 3D T-pose representation $\mathcal{G}$ includes two sets of Gaussians represented as splatter images [22]: $\mathcal{G}^T = \{\mathbf{G}_{ij}^T\}_{i=1j=1}^{H^T W^T}$ from the template branch and $\mathcal{G}^I = \{\mathbf{G}_{n,ij}^I\}_{n=1i=1j=1}^{N \ H^I \ W^I}$ from the image branches. We will provide details regarding the two branches in Sec. 3.1. $H^T, W^T$ and $H^I, W^I$ are the height and width of the template branch and image branches, respectively. Each element of the splatter image consists of the following components:

$$\mathbf{G}_{\cdot}^* = (\boldsymbol{\mu}_{\cdot}^*, \mathbf{s}_{\cdot}^*, \mathbf{r}_{\cdot}^*, o_{\cdot}^*, \mathbf{h}_{\cdot}^*, \mathbf{w}_{\cdot}^*), \tag{3}$$

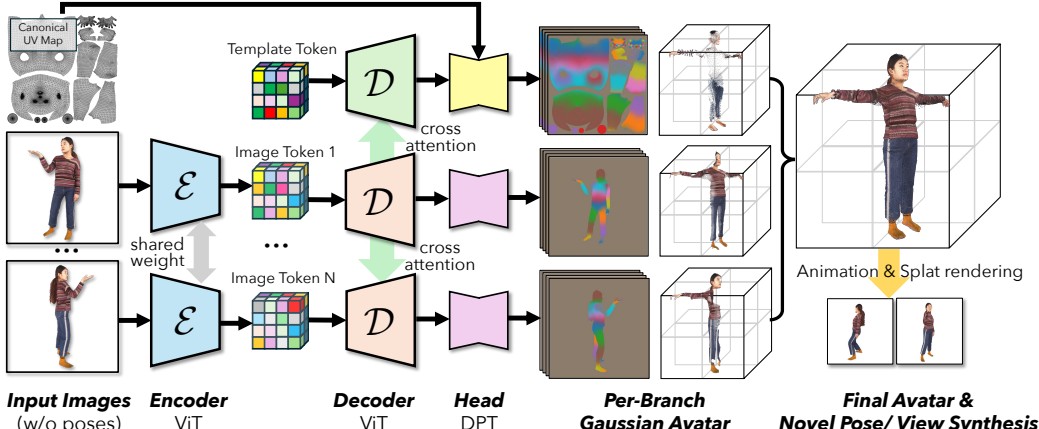

| Input Images | Encoder | Decoder | Head | Per-Branch | Final Avatar & |
| (w/o poses) | ViT | ViT | DPT | Gaussian Avatar | Novel Pose/ View Synthesis |

Figure 2: **Model architecture of the reconstruction module.** The reconstruction module reconstructs the canonical T-pose representation solely from images. It follows the encoder-decoder structure and consists of two types of branches: a template branch and image branches. We show two views of the predictions of each branch: the splatter images in the 2D format and their visualizations in 3D. Gaussians predicted from all branches are combined and fed into the articulation and rendering.

which refers to either $\mathbf{G}_{ij}^T$ or $\mathbf{G}_{n,ij}^I$, and subsumes a set of variables: $\boldsymbol{\mu}_{\cdot}^* \in \mathbb{R}^3$ is the mean of the Gaussian at location "·" in branch "*", $\mathbf{s}_{\cdot}^* \in \mathbb{R}^3$ and $\mathbf{r}_{\cdot}^* \in \mathbb{R}^4$ are the scale and rotation represented via quaternions, $o_{\cdot}^*$ is the opacity, $\mathbf{h}_{\cdot}^* \in \mathbb{R}^{\mathrm{Deg} \times 3}$ refers to the spherical harmonics of degree Deg, $\mathbf{w}_{\cdot}^* \in \mathbb{R}^O$ denotes the LBS weights assigned to the bones with $O$ being the number of bones.

We detail the reconstruction module in Sec. 3.1, the articulation and rendering module in Sec. 3.2, and we describe the training losses in Sec. 3.3.

## 3.1 Reconstruction

The reconstruction module (Recon), defined in Eq. (2), reconstructs the Gaussian primitives $\mathcal{G}$ in the canonical T-pose space given $N$ input images $\{\mathbf{I}_n\}_{n=1}^N$ and the corresponding subject masks $\{\mathbf{M}_n\}_{n=1}^N$. To achieve this we use the architecture illustrated in Fig. 2. Inspired by DUSt3R and its follow-ups [26, 34, 23], we adopt an encoder-decoder architecture: the encoders embed each input into tokens independently, and the decoders exchange information between inputs. Lastly, we use a regression head to predict the Gaussian primitives from the tokens.

Our model consists of two types of branches: the *template branch* and $N$ *image branches*, one for each input image $\mathbf{I}_n$, with $n \in \{1, \dots, N\}$. Abstractly, the template branch captures the overall structure of the human body and reconstructs the unseen regions of the subject, while the image branches predict pixel-aligned Gaussians that focus only on the visible regions in the input images, yielding a fine-grained but incomplete reconstruction.

**Encoder:** The *template encoder* is intended to capture articulated human-shape knowledge agnostic to subjects identity —here, the average SMPL-X template in T-pose. To this end, we simply implement it as a learnable embedding map $\mathbf{F}_0^T \in \mathbb{R}^{H_0^T \times W_0^T \times C}$, where $H_0^T$ and $W_0^T$ denote the embedding's height and width, and $C$ its channel dimension. Since this embedding takes no input, it is identical for all subjects and independent of the input images. The $N$ *image encoders* share weights and embed each input image $\mathbf{I}_n$ into tokens with a ViT-based architecture, i.e., $\mathbf{F}_{n,0}^I = \mathrm{Enc}^I(\mathbf{I}_n) \in \mathbb{R}^{H_0^I \times W_0^I \times C}$, with $n \in \{1, \dots, N\}$.

**Decoder:** The ViT-based decoders exchange information across all branches. Formally, let $\mathbf{F}_b^T$ denote the tokens in the template branch, and let $\mathbf{F}_{n,b}^I$ refer to those in the $n$-th image branch, while $b \in \{1, \dots, B\}$ indicates the $b$-th decoder block. We compute the tokens as follows: At the $b$-th decoder stage, the decoder block updates each feature—either $\mathbf{F}_b^T$ or $\mathbf{F}_{n,b}^I$—by cross-attending its own feature to all other features from stage $b-1$. Information across images and between images and the template is exchanged and implicitly aligned—this is key to enabling pose independence and allowing the template to "inpaint" missing content in unseen regions. Note, we employ separate

decoder blocks for the template and image branches, however, all $N$ image-branch decoders are multi-layer transformers with shared weights.

**Prediction head:** The last modules are the DPT-based [17] prediction heads. As before, we use different heads for the template branch and the $N$ image branches. In the template branch, the network predicts residuals relative to the SMPL-X T-pose template in the average shape. When added to the rasterized template in UV space, this yields $\mathcal{G}^T$ in Eq. (2). The image branches are rather straightforward: in the $n$-th image branch, we directly predict the Gaussian primitives $\{\mathbf{G}^I_{n,ij}\}^{H^I\,W^I}_{i=1\,j=1}$ in Eq. (2), except that we multiply the predicted opacities by the input subject mask to exclude the background pixels from rendering. Here, $H^I$ and $W^I$ are the height and width of the *input* image. That is, we predict one Gaussian for each foreground pixel in the input image, ensuring that all visible fine-grained details are captured by the representation. Note that the set of Gaussians in the image branches $\mathcal{G}^I$ is in the canonical space (T-pose), not the space of the input poses, making it independent of the human poses in the input images.

Finally, we merge the predicted template and image Gaussians into our final reconstructed avatar by taking the union of the two sets: $\mathcal{G} = \mathcal{G}^T \cup \mathcal{G}^I$.

## 3.2 Articulation and Rendering

We now describe how to render the target image using our recovered avatar $\mathcal{G}$, given target extrinsics $\mathbf{E}$, intrinsics $\mathbf{K}$, and human shape and pose $\mathbf{P} = (\boldsymbol{\beta}, \boldsymbol{\theta})$. We use two steps: *articulation* and *splat rendering*. In the articulation step, we warp the canonical T-posed Gaussian avatar $\mathcal{G}$ into an articulated Gaussian under the target pose via linear blend skinning, i.e., $\mathcal{G}_\mathbf{P} = \texttt{LBS}(\mathcal{G}; \mathbf{P})$, where the LBS weights for each Gaussian are part of the reconstruction module's output (shown in Eq. (3)). Given the warped $\mathcal{G}_\mathbf{P}$, we render the image and its mask via Gaussian splatting, $(\mathbf{I}, \mathbf{M}) = \texttt{SplatRender}(\mathcal{G}_\mathbf{P}; \mathbf{E}, \mathbf{K})$. Hence, the $\texttt{Render}$ function defined in Eq. (1) can be written as: $\texttt{Render}(\mathcal{G}; \mathbf{E}, \mathbf{K}, \mathbf{P}) = \texttt{SplatRender}(\texttt{LBS}(\mathcal{G}; \mathbf{P}); \mathbf{E}, \mathbf{K})$. We discuss the details of this stage and the justification of the design choices in the appendix.

## 3.3 Training losses

The reconstruction and rendering modules are end-to-end trainable. During training, we assume that we have the corresponding camera poses and human poses for the target ground-truth images, which are only used for rendering during training. Neither at training-time nor at test-time does reconstruction depend on any poses of the input images.

Our training loss is

$$L = L_{\text{mse}} + \alpha_{\text{lpips}} L_{\text{lpips}} + \alpha_{\text{chamfer}} L_{\text{chamfer}} + \alpha_{\text{proj}} L_{\text{proj}} + \alpha_{\text{lbs}} L_{\text{lbs}}, \qquad (4)$$

where $\alpha_*$ are hyper-parameters.

$L_{\text{mse}}$ and $L_{\text{lpips}}$ are the MSE loss and the LPIPS loss between the rendered image $\mathbf{I}$ and the ground truth $\mathbf{I}_{\text{gt}}$. $L_{\text{chamfer}}$ is the chamfer distance between the Gaussians from the template branch and the image branches $\mathcal{G}^T$ and $\mathcal{G}^I$.

We use two additional losses. First, the *projection loss* $L_{\text{proj}}$ encourages each image-branch Gaussian can well explain its corresponding image. It consists of two terms: 1) MSE and LPIPS between the $n$-th input image and the rendering using only Gaussians from image branches. 2) $\ell_2$ distance between each pixel $(i, j)$ and the projected mean of its predicted Gaussian $\mathbf{G}^I_{n,ij}$: $\left\| (i, j) - \texttt{Project}\left(\texttt{LBS}(\boldsymbol{\mu}^I_{n,ij}, \mathbf{w}^I_{n,ij}, \mathbf{P}); \mathbf{K}, \mathbf{E}\right) \right\|^2_2$. Here, $\texttt{LBS}(\boldsymbol{\mu}; \mathbf{w}, \mathbf{P})$ warps $\boldsymbol{\mu}$ via linear blend skinning with weights $\mathbf{w}$ under pose $\mathbf{P}$, and $\texttt{Project}(\boldsymbol{\mu}; \mathbf{K}, \mathbf{E})$ projects the 3D point $\boldsymbol{\mu}$ to 2D using intrinsics $\mathbf{K}$ and extrinsics $\mathbf{E}$. Second, the *LBS loss* $L_{\text{lbs}}$, which encourages the predicted LBS weights $\mathbf{w}^*$ in Eq. (3) to follow the pseudo LBS weights obtained from the SMPL-X mesh provided in the training data. We call the LBS weights "pseudo" as they are rasterized from the SMPL-X mesh without clothes and are not strictly aligned with the clothed bodies.

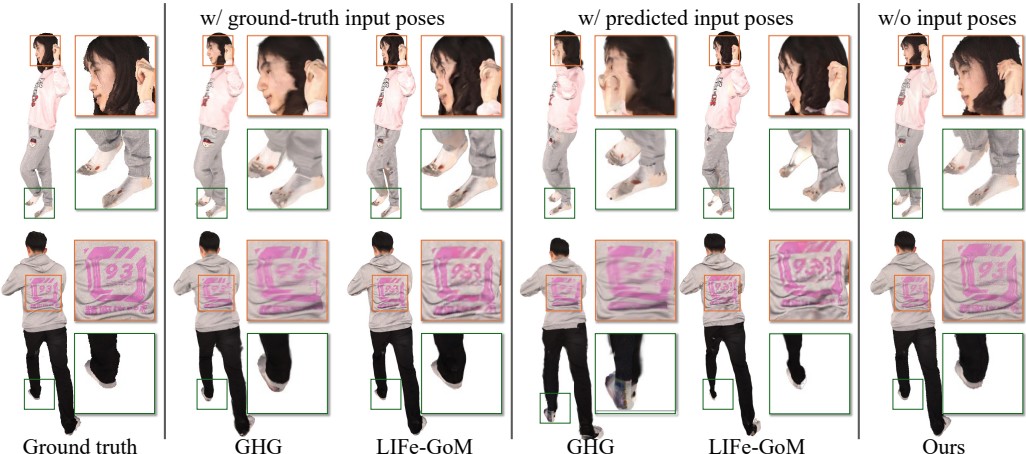

| w/ ground-truth input poses | | w/ predicted input poses | | w/o input poses |

| Ground truth | GHG | LIFe-GoM | GHG | LIFe-GoM | Ours |

Figure 3: **Novel view synthesis from sparse input images on THuman2.0.** Our approach performs on par with the state-of-the-art in the lab setting (with ground-truth input poses in the reconstruction phase in test-time). Sometimes, ours even captures sharper details. In the real setting (with predicted input poses in the reconstruction in test-time), the rendering quality of GHG and LIFe-GoM is largely decayed. However, our approach without pose priors does not suffer from the bad poses.

# 4 Experiments

## 4.1 Experimental setup

**Datasets.** We train our model on THuman2.0 [35], THuman2.1 [35], and HuGe100K [37]. For evaluation purposes, we adopt THuman2.0, HuGe100K and XHuman [21]. We follow GHG's split [10] on THuman2.0. On HuGe100K, we use the scripts provided by IDOL [37] and split each directory into 10 validation subjects, 50 test subjects and the rest for training. Please see the appendix for details.

**Baselines.** For novel view synthesis from sparse view images we compare to GoMAvatar [27], 3DGS-Avatar [15], iHuman [13], NHP [8], NIA [9], GHG [10], and LIFe-GoM [28] on THuman2.0. We use three input images in this task. Note that all baselines use camera poses and human poses as input for reconstruction in test-time, while we do not. Among the baselines, GoMAvatar, 3DGS-Avatar and iHuman require to be optimized for each subject, while others and our method are generalizable. We evaluate two test-time settings: 1) Use of human poses predicted by the state-of-the-art SMPL-X prediction model MultiHMR [1] for test-time reconstruction, which is a realistic real-world setup; 2) Use of ground truth human poses provided by the dataset for test-time reconstruction, which is not a realistic real-world setup, but nonetheless insightful. The target rendering poses in test-time are always ground-truths from THuman2.0. We also evaluate on the XHuman dataset for cross-domain novel view synthesis and novel pose synthesis, which we defer to the appendix due to the page limit. For the task of novel view synthesis from a single image, we compare to IDOL [37] and LHM [16] on HuGe100K. Like our approach, both methods eliminate the need for input poses during the reconstruction phase at test-time. We train IDOL and our method with the same training set on HuGe100K. Since LHM does not open-source the training script and the training dataset, we are unable to compare fairly. Hence, we directly assess the released checkpoints on HuGe100K's test set.

**Evaluation metrics.** We assess PSNR, LPIPS$^*$, and FID following GHG [10] on THuman2.0 [35], XHuman [21], and HuGe100K [37]. We additionally compare the reconstruction time, i.e., the time spent to compute the 3D representation given input images.

## 4.2 Comparisons with baselines

**Novel view synthesis from sparse images.** We evaluate novel view synthesis from sparse images on the THuman2.0 dataset in Tab. 1, using three input views. Other than input images and subject masks, for reconstruction at test-time, GHG and LIFe-GoM also use the predicted SMPL-X poses from MultiHMR [1], the state-of-the-art SMPL-X predictor. For rendering, all methods use the ground-

Table 1: **Comparisons to novel view synthesis from sparse input images on THuman2.0.** All the baselines take the *predicted* poses in reconstruction in test-time from MultiHMR [1] as inputs. We compare in the setting of no test-time optimization, only optimizing the camera poses, and optimizing both camera poses and human poses in test time. Our approach, without pose priors, does not suffer from the errors in the predicted poses. Therefore, ours significantly outperforms the baselines.

| | w/o test-time optim. | | | w/ test-time cam pose optim. | | | w/ test-time all pose optim. | | |
|---|---|---|---|---|---|---|---|---|---|
| | PSNR↑ | LPIPS*↓ | FID↓ | PSNR↑ | LPIPS*↓ | FID↓ | PSNR↑ | LPIPS*↓ | FID↓ |
| GHG | 16.96 | 185.67 | 71.46 | 19.50 | 160.52 | 70.37 | - | - | - |
| LIFe-GoM | 19.70 | 146.19 | 63.34 | 20.52 | 142.53 | 62.78 | 22.59 | 130.71 | 60.74 |
| Ours | **22.49** | **105.45** | **42.19** | **22.94** | **103.94** | **42.25** | **25.33** | **92.32** | **39.66** |

Table 2: **Comparisons on novel view synthesis from sparse input images in reconstruction in test-time on THuman2.0.** All the baselines take the *ground-truth* poses as inputs, which is an unrealistic setting in real-world applications. We use three images as inputs. We compare to baselines w/o test-time pose optimization in the first block and w/ test-time pose optimization in the second block, where we mark the methods with stars. Our approach, without any pose priors, achieves comparable PSNR and better LPIPS* and FID than baselines with ground-truth input poses.

| | Method | PSNR↑ | LPIPS*↓ | FID↓ |
|---|---|---|---|---|
| | GoMAvatar [27] | 23.05 | 133.98 | 87.51 |
| | 3DGS-Avatar [15] | 21.25 | 160.48 | 157.21 |
| | iHuman [13] | 22.77 | 131.67 | 101.70 |
| w/o test-time | NHP [8] | 23.32 | 184.69 | 136.56 |
| optimization | NIA [9] | 23.20 | 181.82 | 127.30 |
| | GHG [10] | 21.90 | 133.41 | 61.67 |
| | LIFe-GoM [28] | **24.65** | 110.82 | 51.27 |
| | Ours | 22.49 | **105.45** | **42.19** |
| w/ test-time | LIFe-GoM* [28] | **25.87** | 108.67 | 50.78 |
| optimization | Ours* | 25.33 | **92.32** | **39.66** |

truth camera poses and human poses provided by the dataset. Due to potential misalignments in scale and human pose between the predicted poses (used during reconstruction) and the ground-truth poses (used for rendering), we further conduct test-time pose optimization. Specifically, we optimize the *target rendering* camera and human poses while keeping the reconstructed representation fixed. This procedure is only used for evaluation purposes. Notably, since GHG is designed solely for novel view synthesis and lacks support for animation, it cannot accommodate changes in human pose during test-time optimization. Accordingly, we report three sets of numbers: w/o test-time pose optimization, w/ test-time camera pose optimization, and w/ test-time camera and human pose optimization. Here, "pose" refers to the target pose used in rendering. Our approach consistently outperforms GHG and LIFe-GoM by a margin in all evaluation protocols. We decrease LPIPS* by over 35 points and FID by over 20 points compared to LIFe-GoM. We further compare with the baselines qualitatively in Fig. 3. When the input poses are insufficiently accurate, the baseline methods struggle to reconstruct high-fidelity details, particularly in fine-grained regions such as the face and feet. In contrast, our approach remains unaffected.

To complete the quantitative comparison, we also report the standard evaluation setting on THuman2.0 [10, 28] in Tab. 2: poses for reconstruction and rendering are both ground truths provided by the dataset. Note that this setting is unrealistic in real-world applications, since the ground-truth input poses are not available for reconstruction at test-time and must be predicted by off-the-shelf tools. Even without any pose priors as inputs, our approach improves upon methods that use ground-truth poses in LPIPS* and FID. Our PSNR is slightly worse than the baselines. This is because PSNR focuses more on pixel-level accuracy. The pose priors provide better alignment between the canonical reconstruction and the target poses (same as input poses). Test-time optimization for target poses in rendering can resolve the potential ambiguity in our approach, as shown in the bottom part of Tab. 2. The qualitative comparisons in Fig. 3, shows that our method can 1) capture details better than the baselines, e.g., the prints on the back of the second person; and, aided by the two-branch design, 2) improves inpainting of unseen regions, e.g., the pants in the first example.

Table 3: **Comparisons on novel view synthesis from a single image on HuGe100K.** All methods do not take input poses in the reconstruction phase. Our approach achieves better PSNR, LPIPS* and FID compared to IDOL. We are not able to fairly compare to LHM due to missing training scripts of LHM. However, our reconstruction is much faster than LHM.

| Method | PSNR↑ | LPIPS*↓ | FID↓ | Reconstruction time↓ |
|---|---|---|---|---|
| LHM-500M [16] | 17.48 | 129.63 | 25.65 | 2.69s |
| LHM-1B [16] | 17.48 | 128.85 | 24.72 | 7.41s |
| IDOL [37] | 20.89 | 111.68 | 16.91 | **311.89ms** |
| Ours | **23.15** | **90.63** | **15.56** | 321.58ms |

In terms of the reconstruction time, our method needs 1.32s on an NVIDIA A100 when processing three input images of resolution $1024 \times 1024$. This is slightly slower than LIFe-GoM's 908ms, but still faster than the fastest per-scene optimization approach, iHuman, which requires more than 7s.

**Novel view synthesis from a single image.** For novel view synthesis from a single image we compare to two recent works, IDOL [37] and LHM [16]. All three methods are similar in three aspects: 1) They do not require poses as input for reconstruction; 2) They need SMPL-X shape and pose for rendering; 3) The reconstructions are in a pre-defined canonical pose. We split HuGe100K [37] into training, validation and test set using the scripts provided by IDOL. We report the performance on the test set in Tab. 3 and qualitative results in Fig. 4. We use the same training setting for IDOL and our method. Since LHM training code wasn't available prior to the submission, we are unable to conduct a fair comparison. Therefore, we directly apply the pretrained weights provided by LHM to HuGe100K's test set. Our method improves upon IDOL, especially in PSNR and

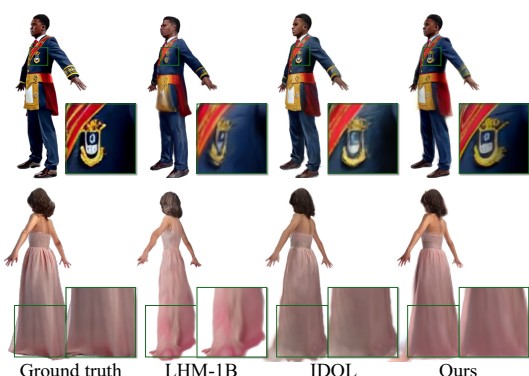

Figure 4: **Comparisons on novel view synthesis from a single image on HuGe100K.** Our model details better than IDOL and LHM. Meanwhile, it can also reconstruct the challenging clothes, such as long dresses.

LPIPS*. Benefitting from the image branch, our method captures observed details better. To verify, we assess results on HuGe100K data separately for all-observed views (front-facing) and for views with little/no overlap with the input image (three back views, camera rotated by at least 120 degree) in Tab. 4. The larger gain in all-observed views than views with little/no overlap with the input image (in LPIPS*, -38.72 vs. -19.37 over IDOL, -56.35 vs. -37.84 over LHM) highlights efficacy of our image branch in capturing observed details. We also measure the reconstruction time, i.e., the time taken to compute the canonical representation. Our method uses 321.58ms, taking a single image of resolution $896 \times 640$ as input. This is similar to IDOL, and much faster than LHM.

We further conduct the experiment of the single-view setting on THuman2.0 in Tab. 5. We use white background following IDOL's default setting. Same as on HuGe100K, we train our approach and IDOL on THuman2.0 while not finetuning LHM. Our approach significantly outperforms IDOL and LHM on PSNR, SSIM and LPIPS*.

### 4.3 Ablation studies

We demonstrate the effectiveness of the key design choices. We use THuman2.1 as the training set and 100 held-out subjects from THuman2.0 as the test set in this section.

**Ablations on the template branch and image branches.** Our model consists of two branches: a single template branch that injects prior knowledge about the human and inpaints the missing parts, and $N$ image branches that output pixel-aligned Gaussians to ensure the fine-grained details observed in the input images are represented. We demonstrate the importance of both types of branches in Tab. 6 and Fig. 5(left). When input images are very sparse, e.g., one single input image ($N = 1$), the template branch plays a vital role in inpainting large unseen regions. This is shown in the first block of Tab. 6: the model with only image branches performs the worst, especially in LPIPS* and FID, as it fails to inpaint the missing regions; the model with only a template branch yields results similar to

Table 4: **Comparisons on novel view synthesis from single image dividing views to all-observed views and views with little/no overlap with the input image on HuGe100K.** We split the test views by their overlap with the input image. We evaluate approaches on all-observed views (front-facing, same view as the input images) and views with little/no observation (three back views, camera rotated by at least 120 degree from the input views). Compared to the baselines, our approach gains larger in all-observed views than views with little/no overlap with the input image highlights efficacy of our image branch in capturing observed details.

| Method | All-observed views | | | Views with little/no observation | | |
|---|---|---|---|---|---|---|
| | PSNR↑ | LPIPS*↓ | FID↓ | PSNR↑ | LPIPS*↓ | FID↓ |
| LHM-1B [16] | 18.56 | 105.86 | 23.25 | 17.17 | 138.34 | 37.27 |
| IDOL [37] | 23.16 | 88.23 | 20.30 | 20.38 | 119.87 | 26.99 |
| Ours | 26.64 | 49.51 | 14.81 | 22.53 | 100.50 | 24.14 |

Table 5: **Comparisons on novel view synthesis from a single image on THuman2.0.** All methods do not take input poses in the reconstruction phase. Our approach achieves better PSNR, LPIPS* and FID compared to IDOL.

| Method | PSNR↑ | LPIPS*↓ | FID↓ |
|---|---|---|---|
| LHM-1B [16] | 22.03 | 70.31 | 63.34 |
| IDOL [37] | 23.47 | 66.62 | 83.37 |
| Ours | **24.64** | **49.69** | **34.82** |

the model with all branches. When we increase the number of input images to 3, the image branches model most of the regions in better detail. Using the template branch only fails to capture the fine textures, hence performing the worst in FID. The model with both types of branches attains the best overall results across different numbers of inputs.

**Ablations on auxiliary losses $L_{\text{proj}}$ and $L_{\text{lbs}}$ in Eq.** (4). We report the quantitative results in Tab. 7. The projection loss $L_{\text{proj}}$ encourages the image branches to predict pixel-aligned Gaussians so that the observed details are modeled. Without the project loss, the image branches do not output any visible Gaussians. Only the coarse template Gaussians are used in the rendering, which results in missing fine-grained textures, as shown in Fig. 5(middle).

Our approach reconstructs the avatar in the canonical T-pose, different from the poses in the input images. Together with the predicted linear blend skinning weights, this reconstruction can be easily animated to novel poses. Removing $L_{\text{lbs}}$ does not affect novel view synthesis, but the image branches sometimes reconstruct in the pose shown by the input images instead of the canonical T-pose. Further, the model cannot learn appropriate LBS weights, affecting novel pose synthesis. We visualize the learned LBS weights and the canonical space of the corresponding image branch in Fig. 5(right).

$L_{\text{chamfer}}$ in Eq. (4) speeds up convergence of the image branches at the beginning of the training, but we do not find it useful in the final model.

## 4.4 Zero-shot downstream tasks

In addition to novel view and pose synthesis, our model can potentially generalize to downstream tasks in a zero-shot manner—that is, without being explicitly trained for any of them.

**Part segmentation.** We predict the pixel-aligned linear blend skinning weights together with other Gaussian primitives in the image branches. The weights can be converted to segmentation masks, which indicate the body part that a pixel belongs to. In the second column of Fig. 6, we showcase the parts in different colors.

Figure 6: **Zero-shot downstream tasks.** Our model can be adapted to part segmentation and pose estimation without finetuning.

Table 6: **Ablations on the template branch and image branches.** Taking $N = 1$ or $N = 3$ input images, we train and evaluate our approach with the template branch only, image branches only and both types of branches. Using both types of branches offers the best performance across different numbers of input images.

Table 7: **Ablations on auxiliary losses $L_{\text{proj}}$ and $L_{\text{lbs}}$.** We compare three training losses: (a) w/o $L_{\text{proj}}$, (b) w/o $L_{\text{lbs}}$ and (c) w/ all losses. Without $L_{\text{proj}}$, the image branches cannot predict pixel-aligned Gaussians. Without $L_{\text{lbs}}$, the image branches fail to reconstruct in the canonical T-pose.

| $N$ inputs | Branch types | PSNR↑ | LPIPS*↓ | FID↓ |
|---|---|---|---|---|
| 1 | Template branch only | 21.36 | 121.46 | 56.01 |
|  | Image branches only | 21.51 | 134.96 | 66.53 |
|  | All branches | 21.41 | 124.36 | 57.78 |
| 3 | Template branch only | 22.13 | 108.85 | 47.60 |
|  | Image branches only | 22.03 | 109.23 | 42.05 |
|  | All branches | 22.23 | 106.98 | 42.18 |

| $L_{\text{proj}}$ | $L_{\text{lbs}}$ | PSNR↑ | LPIPS*↓ | FID↓ |
|---|---|---|---|---|
| ✗ | ✓ | 22.09 | 110.37 | 48.98 |
| ✓ | ✗ | 22.18 | 107.28 | 43.17 |
| ✓ | ✓ | 22.23 | 106.98 | 42.18 |

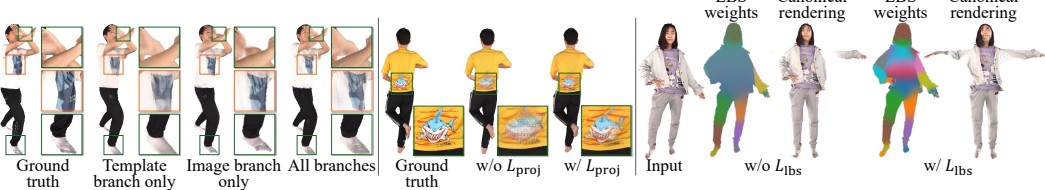

Figure 5: **Ablation studies. Left:** Ablations on the template branch and image branches. Taking a single image as input, template branch only cannot model fine details, such as the prints on the T-shirts (orange boxes). Image branches only miss unseen regions (green boxes). Using both branches offers the best overall quality. **Middle:** Ablation on $L_{\text{proj}}$. Without $L_{\text{proj}}$, only the template Gaussians are effective in the rendering, leading to blurry results. **Right:** Ablation on $L_{\text{lbs}}$. Without supervised with the pseudo LBS weights, the image branch fails to reconstruct in the canonical T-pose and to predict the correct LBS weights.

**Human pose estimation.** In the image branches, we also predict pixel-aligned 3D points in the canonical T-pose. This naturally provides correspondences between the 3D canonical points and the 2D pixel coordinates. Using correspondences, we can optimize the body poses, as shown in the third column of Fig. 6 ("Pose estimation (one stage)"). Since there can be multiple valid poses that satisfy the correspondences, we then adopt a second stage for pose optimization, where we optimize the poses via photometric losses, i.e., we treat $\mathbf{P}$ initialized from the first stage as the optimizable parameters in Eq. (1). The results are shown in the last column of Fig. 6 ("Pose estimation (two stages)"). We highlight the improved poses with orange boxes.

## 5 Conclusions

We propose NoPo-Avatar, a novel model that reconstructs the canonical representation from sparse input images, while not using any human poses. Our model consists of two branches: a template branch that injects prior knowledge about human shapes and inpaints the missing parts, and image branches that predict pixel-aligned Gaussians to depict the fine-grained details. Our approach achieves comparable results to state-of-the-arts which use ground-truth pose priors during test-time reconstruction, and significantly outperforms those that use predicted pose priors during test-time reconstruction.

**Broader Impact.** The proposed method can benefit AR/VR applications. Concerns remain regarding the identity and authenticity of reconstructed avatars. To address these issues, we advocate for the use of strict licensing agreements and responsible usage practices for the proposed methods.

**Acknowledgements.** Work supported in part by NSF grants 2008387, 2045586, 2106825, NIFA award 2020-67021-32799, and OAC 2320345.

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

# Appendix — NoPo-Avatar: Generalizable and Animatable Avatars from Sparse Inputs without Human Poses

The appendix is structured as follows:

- We provide implementation details in Sec. A.
- We elaborate on the baseline setup and experimental details in Sec. B.
- Due to space limitations in the main paper, we clarify the experimental settings and provide additional results in Sec. C.
- We discuss the limitations and future works in Sec. D.

We also provide a webpage `index.html` in the supplementary. It lists videos for freeview rendering comparisons to baselines, novel pose synthesis, and cross-domain generalization.

## A   Implementation Details

### A.1   Articulation

We describe in detail how we articulate the canonical representation into pose $\mathbf{P} = (\boldsymbol{\beta}, \boldsymbol{\theta})$, where $\boldsymbol{\beta}$ is the shape and $\boldsymbol{\theta}$ is the pose represented by the orientation of each joint. We assume that the canonical representation $\mathcal{G}$ follows the skeleton of the average template shape in T-pose, i.e., the skeleton acquired by setting the shape parameters and pose parameters to all zeros in SMPL-X. When warping with pose $\mathbf{P}$, we first reshape the canonical representation consuming the target shape $\boldsymbol{\beta}$, and then rotate the reshaped Gaussians in T-pose by the pose $\boldsymbol{\theta}$.

**Step 1: Reshaping by $\boldsymbol{\beta}$.** People are of various heights and bone lengths. To reshape the canonical representation, we first compute the skeleton of the average shape and the skeleton of shape $\boldsymbol{\beta}$, both in T-pose. Then we calculate the transformation between each pair of corresponding bones in two skeletons, denoted as $\{s_i, R_i, t_i\}_{i=1}^{O}$. Here, $s_i \in \mathbb{R}$ accounts for the scaling factor of the bone length, and $R_i \in SO(3)$ and $t_i \in \mathbb{R}^3$ are the rotation and translation, respectively. $O$ denotes the number of bones. In Eq. (3), we define the LBS weights as $w^* = \{w_{\cdot i}^*\}_{i=1}^{O} \in \mathbb{R}^O$. We then transform the Gaussian's mean $\mu_{\cdot}^*$, and rescale and rotate a Gaussian's covariance:

$$\mu_{\cdot}^{*\boldsymbol{\beta}} = \sum_{i=1}^{O} w_{\cdot i}^*(s_i R_i \mu_{\cdot}^* + t_i), \quad \Sigma_{\cdot}^{*\boldsymbol{\beta}} = \left(\sum_{i=1}^{O} w_{\cdot i}^*(s_i R_i)\right)^T R_{\cdot}^{*T} S_{\cdot}^{*T} S_{\cdot}^* R_{\cdot}^* \left(\sum_{i=1}^{O} w_{\cdot i}^*(s_i R_i)\right). \quad (5)$$

$R_{\cdot}^{*T} S_{\cdot}^{*T} S_{\cdot}^* R_{\cdot}^*$ is the original covariance matrix, where $R_{\cdot}^*$ and $S_{\cdot}^*$ are the matrix form of the predicted rotation and scale of the Gaussian. We rescale the covariance matrices accordingly to avoid artifacts due to reshaping.

**Step 2: Rotating by $\boldsymbol{\theta}$.** $\mu_{\cdot}^{*\boldsymbol{\beta}}$ and $\Sigma_{\cdot}^{*\boldsymbol{\beta}}$ are still in T-pose. We then warp it by pose $\boldsymbol{\theta}$, which represents the orientations of each *joint*. This step is a standard linear blend skinning. However, we need to convert the predicted LBS weights associated with bones $w^* \in \mathbb{R}^O$ to weights associated with joints. As the skeleton is defined in a tree structure, each bone connects a parent joint and a child joint. We simply assign the LBS weights associated with the bone to its parent joint. We illustrate this choice in Fig. 7.

Alternatively, the reconstruction module could directly predict the canonical representation in the target shape, thereby eliminating the need for Step 1 in the articulation. However, we find that decoupling the shape from the canonical representation and retaining Step 1 is empirically important. Due to the decoupling, the canonical representation $\mathcal{G}$ from the reconstruction stage follows a fixed skeleton, i.e., the skeleton of the average template shape in T-pose, which alleviates the scale ambiguity inherent in this task. We show the artifacts without decoupling the shape from $\mathcal{G}$ and Step 1 in Fig. 8(middle) and the improved rendering quality with our design choice in Fig. 8(right).

At test-time, we use the predicted or user-specified shape and pose as $\boldsymbol{\beta}$ and $\boldsymbol{\theta}$, similar to IDOL [37] and LHM [16]. These parameters are only used in the articulation and rendering stage, not the reconstruction stage. Importantly, our approach remains pose-free in the reconstruction stage.

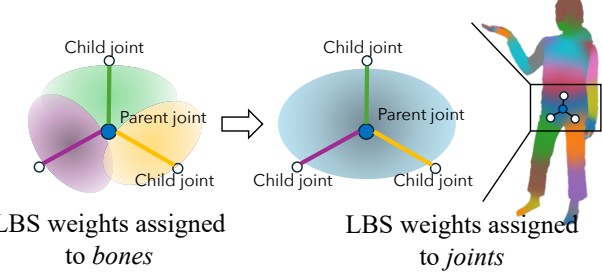

LBS weights assigned to *bones* → LBS weights assigned to *joints*

Figure 7: **Converting LBS weights assigned to bones to LBS weights assigned to joints.** We illustrate the LBS weights around the pelvis root. Our model predicts the LBS weights corresponding to the three bones connected to the pelvis root. The pelvis root serves as the parent joint of three child joints. The LBS weights of the three bones (left) are aggregated and reassigned to the pelvis root (right).

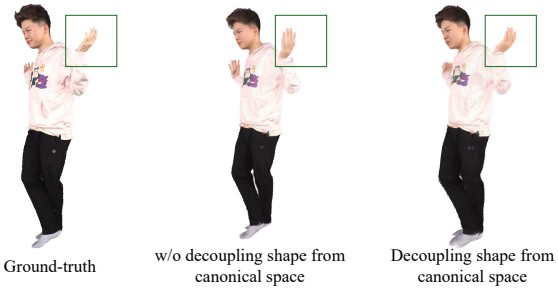

Ground-truth     w/o decoupling shape from canonical space     Decoupling shape from canonical space

Figure 8: **Decoupling the shape from the canonical representation.** Without the decoupling (middle), the model sometimes fails to reconstruct the thin structures, e.g., hands. Decoupling the shape from the canonical representation eases the reconstruction ambiguity.

## A.2 Training and Inference Details

The learnable embedding in the template encoder $F_0^T$ is in the shape of $16 \times 16 \times 1024$. We use $\alpha_{\text{lpips}} = 0.05$, $\alpha_{\text{chamfer}} = 0.1$, $\alpha_{\text{proj}} = 1.0$, and $\alpha_{\text{lbs}} = 0.01$ in Eq. (4).

The model is pretrained from NoPoSplat [34]. To reduce training time, we first train using a low-resolution template and images. The model is first trained at a resolution of $256 \times 256$ for 300K iterations, then upsampled to $512 \times 512$ for another 300K iterations, and finally fine-tuned at full resolution for an additional 50K iterations. We train for 80K iterations in the full resolution of $896 \times 640$ on HuGe100K.

We use a batch size of 4 for all experiments on THuman2.0, THuman2.1, and THuman2.1 + HuGe100K. The training process takes roughly 12 days, 4 days for each stage. For the comparison to LHM and IDOL on HuGe100K, we use a batch size of 16 in the first two stages and 8 in the last stage.

We use two NVIDIA L40S for training at a resolution of $256 \times 256$, four NVIDIA L40S for $512 \times 512$ resolution training, and four NVIDIA H200 for the last stage, i.e., for full resolution training. Inference, including both reconstruction and rendering, requires approximately 24GB of GPU memory for three input images at a resolution of $1024 \times 1024$, and around 11GB for a single image at $896 \times 640$.

## A.3 Zero-shot Human Pose Estimation

We show that our model can perform human pose estimation via analysis-by-synthesis in a zero-shot manner in Sec. 4.4. For this, we use a two-stage optimization process.

**Stage 1: Optimization with projection losses.** Recall that the image branches in our model predict pixel-aligned Gaussians, which means that we obtain one Gaussian with mean $\mu^I_{n,ij}$ and LBS weights $w^I_{n,ij}$ for each pixel $(i,j)$ in the $n$-th image branch. We optimize for the human shape and pose $\mathbf{P}$ by minimizing $\left\|(i,j) - \texttt{Project}\big(\texttt{LBS}(\mu^I_{n,ij}, \mathbf{w}^I_{n,ij}, \mathbf{P}); \mathbf{K}, \mathbf{E}\big)\right\|^2_2$ summed over all foreground pixels, where $\texttt{LBS}(\mu; \mathbf{w}, \mathbf{P})$ warps $\mu$ via linear blend skinning with weights $\mathbf{w}$ under pose $\mathbf{P}$, and $\texttt{Project}(\mu; \mathbf{K}, \mathbf{E})$ projects the 3D point $\mu$ to 2D using intrinsics $\mathbf{K}$ and extrinsics $\mathbf{E}$. We set $\mathbf{E}$ to the identity matrix and let $\mathbf{K}$ have fixed focal lengths and principal points.

**Stage 2: Optimization with photometric losses.** There can be multiple valid poses that satisfy the correspondences. Meanwhile, the reconstruction from image branches is sometimes unreliable for fine-grained structures. We therefore optimize with a photometric loss in the second stage. Specifically, we treat $\mathbf{P}$ as the learnable parameter, initialized by the result of the first stage. We warp the Gaussians by $\mathbf{P}$ and fixed LBS weights, render the images, and compute the photometric loss between the rendered images and the ground-truth, which is the image we estimate the pose from in pose estimation.

As is shown in Sec. 4.4, optimizing with projection losses solely already leads to good results. This is a side benefit from our design choice of predicting pixel-aligned Gaussians from the image branches. Adding the second stage which optimizes with photometric losses further improves the estimation. Note that optimizing with photometric losses alone does not lead to meaningful results, since it requires a good initialization.

# B  Experimental Details

## B.1  Datasets

**THuman2.0.** THuman2.0 provides 526 3D scans as well as the corresponding SMPL-X parameters. We follow GHG [10] to render them into 64 multiview images in the resolution of $1024 \times 1024$ and split them into 426 subjects for training and 100 for evaluation. THuman2.0 uses a special license agreement[1], which we follow.

**THuman2.1.** THuman2.1 extends THuman2.0 to $\sim$2500 scans. Note that the 526 scans from THuman2.0 are included in THuman2.1. The newly added subjects are combined with THuman2.0's 426 training subjects as the new training set. The license agreement is the same as THuman2.0.

**XHuman.** XHuman consists of 20 subjects. It provides the 3D scans and corresponding SMPL-X poses of multiple motion sequences for each subject. We follow LIFe-GoM [28] to evaluate the cross-domain generalization in novel view synthesis on XHuman. The pose estimator MultiHMR fails in one sample so that we cannot get the estimated input poses for the reconstruction phase of GHG and LIFe-GoM. So we remove that sample from the test set. We additionally evaluate novel pose synthesis on the XHuman dataset. For each subject, images from the first frame of a sequence are used as input. The two sequences designated as test sequences are selected as target poses for novel pose synthesis. For each target pose, we render images from three different views. XHuman uses a special license agreement[2], which we follow.

**HuGe100K.** HuGe100K is a synthetic dataset that contains more than 100K subjects. For each subject, it provides a video of 360-degree freeview rendering, camera poses of each frame and the SMPL-X parameters. We use SAM [6] to acquire the subject masks. The dataset groups the subjects into directories. We follow the scripts provided by IDOL [37] and split each directory into 10 validation subjects, 50 test subjects and the rest for training. We use frame 19 as the input view and sample 6 views as the target views for evaluation. HuGe100K uses DeepFashion's license[3], which we follow.

## B.2  Pose Estimation

The baselines, including GHG [10] and LIFe-GoM [28], require camera poses and human poses of the input images as input for the reconstruction phase. As mentioned in the paper, the ground-truth

---

[1]https://github.com/ytrock/THuman2.0-Dataset?tab=readme-ov-file#agreement
[2]https://xhumans.ait.ethz.ch
[3]https://mmlab.ie.cuhk.edu.hk/projects/DeepFashion.html

poses are typically not available in real applications. Instead, the poses have to be estimated by off-the-shelf tools prior to reconstruction.

We choose MultiHMR [1] as the pose estimator for all our experiments requiring estimated poses. MultiHMR is the state-of-the-art end-to-end pose estimator. Different from RoGSplat [30], we do not use EasyMoCap[4] as the pose estimator, since EasyMoCap requires calibrated camera poses, which are unavailable in real settings. Complex pose estimation pipelines such as the one used in ExAvatar [21] are also not preferred. Though accurate, they take over 20 minutes to acquire the poses from a monocular video, which is impractical in real-world applications.

MultiHMR predicts per-image poses. Since LIFe-GoM can take images with different poses as input, we feed the poses independently predicted from each view as the input poses to LIFe-GoM. GHG, however, must take multiview images with the same pose. Therefore, we use the human poses predicted from one view (most frontal view) as the input human pose and register the other views.

### B.3 Additional Information on Test-time Pose Optimization

We perform test-time optimization of both camera pose and human pose in Tab. 1 and Tab. 2. In Tab. 1, we use predicted poses for GHG and LIFe-GoM during test-time reconstruction. During rendering, we always use the ground-truths from the dataset as the target poses for evaluation. The poses in the reconstruction stage may not align with the poses in the rendering stage. For example, the predicted poses assume a tall person photographed by a faraway camera, while the ground-truth target poses assume a shorter person captured by a closer camera. We want to rule out the potential drop in the evaluation metrics due to such misalignment. Though not taking any poses in the test-time reconstruction, our approach may also have the same misalignment between the reconstruction and the rendering poses. Therefore, we optimize the target camera poses and human poses *only for evaluation purposes*. We similarly apply test-time optimization to our method, as shown in Tab. 2, and also apply it to LIFe-GoM using ground-truth input poses for a fair comparison.

## C Addtional Experiments

We present additional experiments, including cross-domain generalization results, a quantitative analysis of how different types of noise in input poses affect baseline performance during test-time reconstruction, and evidence that injecting noise during training does not necessarily enhance robustness to noisy poses at test time. **Additional qualitative video comparisons are provided in the supplementary material at** `index.html`**.**

### C.1 Cross-dataset Evaluation With Sparse Images as Input

We show results of experiments with different training settings in Tab. 8. We choose three training settings of different scales: 1) THuman2.0 which has 426 subjects; 2) THuman2.1 which has 2345 subjects, around $5\times$ the scale of THuman2.0; 3) THuman2.1 + HuGe100K which has over 100K subjects, over $200\times$ the scale of THuman2.0. Meanwhile, we evaluate on three settings: 1) In-domain evaluation on novel view synthesis: The test set is 100 subjects in THuman2.0. We call it in-domain since the training and test sets are both from THuman2.0. 2) Cross-domain evaluation on novel view synthesis: The test set consists of subjects from XHuman, which differ from the training set. 3) Cross-domain evaluation on novel pose synthesis: The test set is also from XHuman, but we evaluate novel poses.

For both in-domain and cross-domain evaluation, novel view synthesis or novel pose synthesis, ours consistently outperforms LIFe-GoM with predicted input poses.

More importantly, we find that our method benefits from scaling up, especially in the cross-domain evaluation. When switching the training data from THuman2.0 to THuman2.1, our approach improves by 0.6 in PSNR for novel view synthesis and by 0.4 for novel pose synthesis in the cross-domain evaluation. Similar improvements can also be observed in LPIPS* and FID scores: LPIPS* improves from 129.04 to 119.84 and FID improves from 47.58 to 41.55. However, LIFe-GoM, which incorporates more hand-crafted inductive biases, does not exhibit such a scaling ability. Equipped with an even larger dataset, such as HuGe100K, we also observe an improvement in PSNR. But LPIPS*

---

[4]https://github.com/zju3dv/EasyMocap

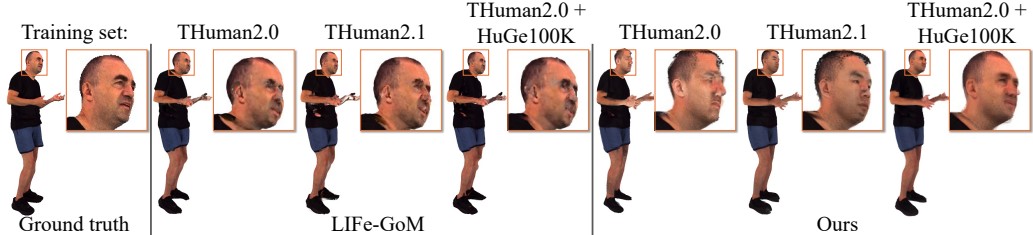

Figure 9: **Novel pose synthesis and cross-domain generalization from sparse input images using different training set sizes.** Our approach scales with the size of the training set, achieving improved identity recovery, whereas LIFe-GoM does not exhibit this behavior.

Table 8: **Comparison on in-domain evaluation and cross-domain generalization using different training set sizes.** We evaluate in-domain novel view synthesis on THuman2.0 and cross-domain novel view synthesis and novel pose synthesis on XHuman. The input poses to LIFe-GoM are *predicted*. Our approach improves with larger-scale training data, while LIFe-GoM does not. The lines labeled with * indicate results w/ test-time pose optimization. We optimize both camera poses and human poses to eliminate the misalignment between the poses used during the reconstruction and the target poses used for rendering. In all settings, ours consistently outperforms LIFe-GoM.

| Training set | THuman2.0 | | | THuman2.1 | | | THuman2.1 + IDOL | | |
|---|---|---|---|---|---|---|---|---|---|
| | PSNR↑ | LPIPS*↓ | FID↓ | PSNR↑ | LPIPS*↓ | FID↓ | PSNR↑ | LPIPS*↓ | FID↓ |
| In-domain evaluation on THuman2.0, novel view synthesis | | | | | | | | | |
| LIFe-GoM | 19.70 | 146.19 | 63.34 | 19.41 | 147.75 | 62.11 | 19.50 | 154.21 | 67.99 |
| LIFe-GoM* | 23.01 | 129.56 | 60.68 | 22.86 | 128.34 | 58.96 | 22.65 | 139.30 | 65.77 |
| Ours | 22.49 | 105.45 | 42.19 | 22.23 | 106.98 | 42.18 | 22.41 | 114.14 | 48.69 |
| Ours* | 25.33 | 92.32 | 39.66 | 25.94 | 87.94 | 38.45 | 25.73 | 97.39 | 46.07 |
| Cross-domain evaluation on XHuman, novel view synthesis | | | | | | | | | |
| LIFe-GoM | 20.94 | 136.42 | 56.02 | 20.50 | 141.24 | 59.94 | 20.60 | 153.43 | 69.54 |
| LIFe-GoM* | 24.39 | 116.25 | 52.19 | 24.22 | 117.07 | 54.43 | 23.91 | 132.16 | 64.47 |
| Ours | 20.96 | 129.04 | 47.58 | 21.56 | 119.84 | 41.55 | 22.52 | 114.95 | 43.59 |
| Ours* | 25.05 | 100.86 | 41.82 | 26.23 | 91.17 | 37.41 | 26.70 | 92.18 | 40.05 |
| Cross-domain evaluation on XHuman, novel pose synthesis | | | | | | | | | |
| LIFe-GoM | 20.95 | 132.86 | 45.22 | 20.49 | 137.62 | 47.69 | 20.56 | 147.52 | 55.05 |
| LIFe-GoM* | 23.98 | 116.42 | 42.64 | 23.70 | 117.81 | 44.37 | 23.56 | 129.09 | 52.01 |
| Ours | 20.93 | 130.89 | 38.65 | 21.36 | 124.96 | 34.67 | 21.74 | 124.91 | 38.27 |
| Ours* | 24.79 | 103.09 | 34.68 | 25.48 | 97.81 | 31.79 | 25.76 | 99.12 | 35.13 |

and FID do not improve further. This is likely because HuGe100K synthesizes multiview images of avatars using diffusion models. The multiview consistency is hence not guaranteed. Inconsistency in the training data may induce blurred rendering, which eventually affects perceptual-based evaluation metrics. We showcase the scaling ability qualitatively in Fig. 9. Since THuman2.0 and THuman2.1 only contain Asian people, our approach may not generalize well to other ethnicities. Our approach benefits from the increasing diversity in HuGe100K and reconstructs better in the cross-domain data. LIFe-GoM, due to hand-crafted inductive biases in its architecture, works better when trained with a small-scale dataset.

## C.2 Comparison of Different Noise Levels in the Input Poses During Test-time Reconstruction

In Fig. 1(a), we show that the baselines including GHG and LIFe-GoM are sensitive to noise in input poses during test-time reconstruction. We achieve this by injecting Gaussian noise or by predicting the input poses. We provide the quantitative results of injecting Gaussian noise or using predicted poses in Tab. 9, matching the results in Fig. 1(a). We also provide a qualitative comparison with LIFe-GoM under noisy input poses in Fig. 10. As the noise level of the input poses increases during test-time reconstruction, LIFe-GoM's performance degrades noticeably. In contrast, our method, which does not rely on pose priors, remains robust and unaffected by such noise.

Table 9: **Comparison of different noise levels in input poses during test-time reconstruction.** GHG and LIFe-GoM take input camera and human poses during test-time reconstruction. We evaluate their robustness to noisy input poses by injecting Gaussian noise into ground-truth poses or by predicting the input poses. For each Gaussian noise level, we average over 5 runs and report the means and standard deviations. Our method, which does not take any pose priors, outputs the same result, regardless of the noise in input poses. The numbers reported in this table are the same as those in Fig. 1(a).

| Noise level Method | No noise | | | std=0.01 | | |
|---|---|---|---|---|---|---|
| | PSNR↑ | LPIPS*↓ | FID↓ | PSNR↑ | LPIPS*↓ | FID↓ |
| GHG | 21.90 | 133.41 | 61.67 | 21.28±0.04 | 136.71±0.25 | 62.06±0.24 |
| LIFe-GoM | 24.65 | 110.82 | 51.27 | 22.95±0.03 | 118.34±0.21 | 52.87±0.08 |
| Ours | 22.51 | 105.85 | 42.37 | 22.51 | 105.85 | 42.37 |

| Noise level Method | std=0.03 | | | std=0.05 | | |
|---|---|---|---|---|---|---|
| | PSNR↑ | LPIPS*↓ | FID↓ | PSNR↑ | LPIPS*↓ | FID↓ |
| GHG | 19.73±0.08 | 150.08±0.54 | 64.99±0.31 | 18.55±0.09 | 163.39±0.70 | 68.45±0.75 |
| LIFe-GoM | 20.90±0.07 | 133.07±0.35 | 57.28±0.26 | 19.73±0.08 | 144.13±0.47 | 63.60±0.76 |
| Ours | 22.51 | 105.85 | 42.37 | 22.51 | 105.85 | 42.37 |

| Noise level Method | pred | | | | | |
|---|---|---|---|---|---|---|
| | PSNR↑ | LPIPS*↓ | FID↓ | | | |
| GHG | 16.96 | 185.67 | 71.46 | | | |
| LIFe-GoM | 19.70 | 146.19 | 63.34 | | | |
| Ours | 22.51 | 105.85 | 42.37 | | | |

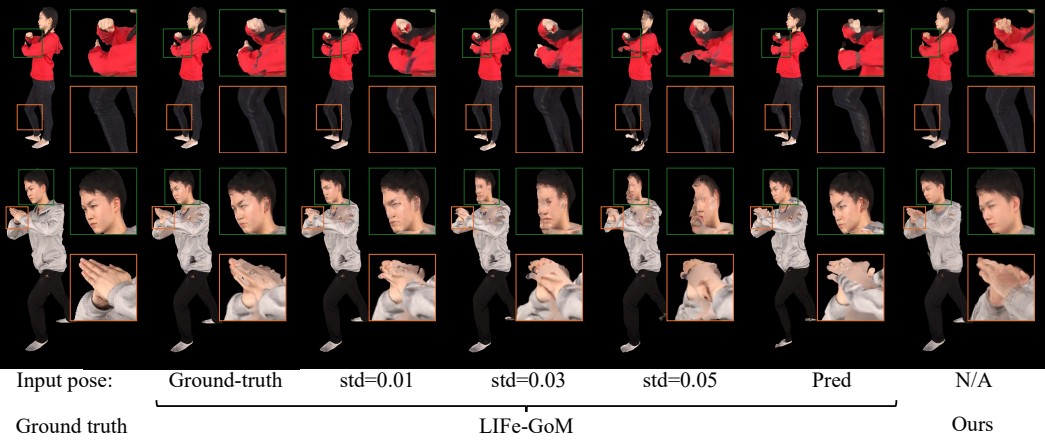

| Input pose: | Ground-truth | std=0.01 | std=0.03 | std=0.05 | Pred | N/A |
|---|---|---|---|---|---|---|
| Ground truth | | | LIFe-GoM | | | Ours |

Figure 10: **Comparison of different noise levels in input poses during test-time reconstruction.** We qualitatively demonstrate the performance degradation of LIFe-GoM under noisy input poses during test-time reconstruction. To simulate noisy conditions, we inject Gaussian noise with varying standard deviations or use predicted poses as inputs. In contrast, our method—relying on no pose priors—remains unaffected by pose noise.

## C.3 Training LIFe-GoM With Noisy Inputs

We train both GHG and LIFe-GoM using ground-truth input poses, but employ predicted poses during test-time reconstruction in the experiments discussed in the main paper and above. This raises a natural question: is the performance drop caused by the gap in input pose quality between training and testing? In this section, we demonstrate that training with noisy input poses does not improve the models' robustness to pose inaccuracies. In fact, training with accurate ground-truth poses often yields the best performance across all levels of input pose noise during test-time reconstruction.

Taking LIFe-GoM as an example, we train it using 1) ground-truth input poses, 2) input poses with synthetic Gaussian noise of std=0.01, 3) input poses with synthetic Gaussian noise of std=0.03, 4)

Table 10: **Training LIFe-GoM with noisy inputs.** We train LIFe-GoM using ground-truth input poses, poses with synthetic Gaussian noise, and predicted poses (pred) during the reconstruction stage. Training with ground-truth poses often yields the best performance across varying levels of pose noise at test time. This indicates that the performance drop observed when using predicted poses during reconstruction is not due to a training–testing gap and cannot be mitigated simply by introducing noise during training.

| | | PSNR↑ | LPIPS*↓ | FID↓ | PSNR↑ | LPIPS*↓ | FID↓ | PSNR↑ | LPIPS*↓ | FID↓ |
|---|---|---|---|---|---|---|---|---|---|---|
| | | | | | | Test input noise | | | | |
| | | | No noise | | | std=0.01 | | | std=0.03 | |
| **Training input noise** | No noise | 24.13 | 110.12 | 49.65 | 22.95 | 118.34 | 52.87 | 20.90 | 133.07 | 57.28 |
| | std=0.01 | 24.11 | 114.97 | 52.87 | 22.74 | 121.70 | 53.85 | 20.67 | 136.17 | 57.36 |
| | std=0.03 | 23.80 | 118.92 | 55.60 | 22.58 | 124.89 | 56.33 | 20.54 | 139.03 | 59.26 |
| | std=0.05 | 23.38 | 122.18 | 57.55 | 22.32 | 127.71 | 58.57 | 20.41 | 141.24 | 61.39 |
| | Pred | 23.38 | 121.66 | 57.34 | 22.32 | 127.44 | 58.22 | 20.65 | 139.44 | 62.38 |
| | | | | | | Test input noise | | | | |
| | | | std=0.05 | | | Pred | | | | |
| **Training input noise** | No noise | 19.73 | 144.13 | 63.60 | 19.70 | 146.19 | 63.34 | | | |
| | std=0.01 | 19.48 | 146.97 | 62.45 | 19.63 | 148.11 | 64.02 | | | |
| | std=0.03 | 19.18 | 151.75 | 62.58 | 19.63 | 150.64 | 66.46 | | | |
| | std=0.05 | 19.10 | 153.60 | 64.42 | 19.54 | 152.07 | 68.48 | | | |
| | Pred | 19.59 | 148.79 | 66.80 | 19.78 | 149.49 | 65.77 | | | |

Table 11: **Robustness to noisy training poses.** We train our approach with noisy training poses. We add synthetic Gaussian noise to the training poses.

| Noise in training poses | PSNR↑ | LPIPS*↓ | FID↓ |
|---|---|---|---|
| No noise | 22.49 | 105.45 | 42.19 |
| std=0.1 | 22.59 | 110.12 | 49.39 |
| std=0.3 | 20.58 | 138.67 | 73.20 |

input poses with synthetic Gaussian noise of std=0.05 and 5) predicted poses from MultiHMR, and evaluate each in all levels of noise. The results are shown in Tab. 10. The more noise we add to the input poses during training, the worse the performance we get across all levels of input pose noise during test-time reconstruction. Hence, the performance drop observed when using predicted poses during reconstruction cannot be mitigated simply by introducing noise during training. Our approach, which takes no pose priors, is an option to eliminate the need for accurate camera and human poses during test time. It is more suitable for real-world applications.

## C.4 Sensitivity to noise in training poses

We use the ground-truth poses provided by the dataset during *training*. To analyze the robustness to poses during training, we add synthetic Gaussian noise. We report the quantitative results in Tab. 11 We find that our approach is robust to some noise, e.g., Gaussian noise of std=0.1: On THuman2.0, it achieves a PSNR/LPIPS*/FID of 22.59/110.12/49.39, compared to 22.49/105.45/42.19 with ground-truth training poses reported in Tab. 2. If we further perturb the ground-truth poses with more significant Gaussian noise of std=0.3, the PSNR/LPIP*/FID drops to 20.58/138.67/73.20. We do not think robustness to the target poses in training is a big concern. Collecting high-quality training data is a one-time effort. Once trained, the model does not rely on poses for reconstruction at inference time. This is different from pose-dependent methods, which still require high-quality input poses during inference for reconstruction.

## C.5 Validating the identity shift

We assess identity shift quantitatively via the facial verification tool DeepFace [19, 20, 18], on HuGe100K. It verifies if two images show the same person. We think this is a good surrogate for identity shift. We adopt all 400 front-facing HuGe100K images and report the verified-rate, i.e., the percentage of "rendered image"-"ground-truth image"-pairs recognized as the same person, and the cosine distance between the rendered face and the ground-truth image in VGG-Face's feature space

Table 12: **Validating the identity shift on HuGe100K.** To quantify the identity shift, we report the verified-rate and the cosine distance between the rendered image and the ground-truth image in VGG-Face's feature space.

| Noise in training poses | Verified-rate↑ | Cosine distance↓ |
|---|---|---|
| LHM-1B | 95.75% | 0.30 |
| IDOL | 93.75% | 0.45 |
| Ours | 95.25% | 0.32 |

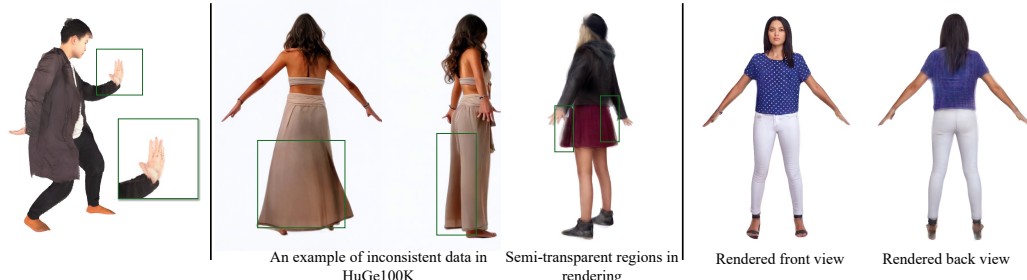

An example of inconsistent data in HuGe100K    Semi-transparent regions in rendering    Rendered front view    Rendered back view

Figure 11: **Limitations. Left:** Incorrect hand geometry. **Middle:** When trained on HuGe100K, our model sometimes produces semi-transparent regions. We suspect the reason is that the data in HuGe100K lack multiview consistency. **Right:** Taking the front view as the only input view, our model renders a sharp front view. But the back view is blurry due to inpainting.

in Tab. 12. Our method without dedicated face module achieves a verified-rate of 95.25% and an average cosine distance of 0.32. LHM-1B has 95.75%/0.30. Note, LHM uses a dedicated face image crop as additional input and a feature pyramid for improved facial modeling. Both ours and LHM outperform IDOL's 93.75%/0.45. We think this shows that our model does not suffer from a severe identity shift as results are comparable to methods with dedicated face module.

## D Limitations

**Failure in modeling expressions and hands.** Our model currently does not support expression retargeting. In most of the input images, hands occupy very few pixels or are heavily occluded. Therefore, predicting the LBS weights and corresponding 3D locations in the canonical T-pose for each pixel in the image branches is sometimes challenging. So the hands are not as sharp as other regions, as is shown in Fig. 11(left). Similar issues happen in LHM. A possible solution is to train two separate models, especially for faces and hands, respectively, compromising the training time and reconstruction speed.

**Sensitivity to inconsistent training data.** Our model sometimes predicts blurry results in unseen regions and semi-transparent regions on the boundary when trained on HuGe100K. Notably, such issues do not happen when training on THuman2.0. We suspect that this is because the HuGe100K data is synthesized by diffusion models. It hence sometimes lacks multiview consistency. We showcase a HuGe100K human subject which is inconsistent in different views in Fig. 11(middle). As a regression-based method, ours is prone to output blurry results, semi-transparent regions, or small floaters on the boundary. A more consistent and higher-quality training set will resolve this issue.

**Blurry inpainting.** Even though our model improves upon LIFe-GoM in inpainting small unseen regions, our model, as a regression-based method, struggles to hallucinate sharp and high-frequency details in large unseen regions, e.g., the dotted pattern on the shirt in Fig. 11(right). Generative models are generally more suitable for hallucinating high-frequency details, which is important for reconstruction and rendering from a single image.

