# OpenReview forum: "NoPo-Avatar: Generalizable and Animatable Avatars from Sparse Inputs without Human Poses"
_NeurIPS.cc/2025/Conference — NeurIPS 2025 poster_

### Official Review · Reviewer_Y9Hx · 2025-06-16

**Clarity:** 3
**Significance:** 3
**Originality:** 2
**Rating:** 4
**Confidence:** 4

**Summary:**

The paper proposes a method to generate human representations using Gaussian primitives in a canonical pose, which are then rendered into images via Gaussian splatting. The approach works with either single or multiple input images and notably removes the need for both human pose and camera pose annotations. Additionally, the generated human representations support animation.

**Questions:**

1) How does the model handle long garments, such as dresses, when predicting residuals in canonical pose? Are there specific mechanisms to account for the complex deformation of such clothing?
2) Could the authors clarify the role of the canonical UV map in the pipeline? Specifically, how is it used, and what advantages does it provide?
3) Does the method generalize well to animating subjects wearing dresses in diverse poses(especially for complex leg motion)? If so, could the authors provide examples or further discussion?

**Ethical Concerns:**

["NO or VERY MINOR ethics concerns only"]

**Final Justification:**

The author has pointed out results for multiview consistency cases, and loose clothes cases reconstruction in the supplementary material. For the missing in-the-wild image evaluation, the author explained that the UBCfashion dataset image is closed to in-the-wild level and referred results in supplementary material applying on these images, however, comparisons are not presented against baseline methods. The function of the UV map in the proposed model has been explained and the limitation of the inpainting ability of the proposed model has been discussed by the author. The rating has been increased based on the discussion.

**Limitations:**

The approach relies on a Gaussian primitive template to inpaint missing content in unseen regions, which could lead to blurry renderings when only a single image or multi-view images with limited coverage of the subject are available. Thus the method performs best when sufficient multi-view coverage is provided, while its effectiveness is reduced in sparse or narrowly-viewed scenarios.

**Quality:**

2

**Strengths And Weaknesses:**

Strengths:
1) The proposed method does not require body pose or camera pose annotations, which simplifies the data requirements and increases its applicability in unconstrained settings.
2) The method provide tools for both body part segmentation and human pose estimation, enhancing the utility of the approach for downstream tasks.
3) The paper is clearly written and well-structured, making the methodology easy to follow.

Weaknesses:
1) The paper does not provide qualitative results on in-the-wild images; all evaluations are conducted on datasets collected in controlled laboratory settings. Given that the method does not require pose information, it would be particularly valuable to demonstrate its performance on in-the-wild data to better assess its robustness and practical applicability.

2) Only a limited number of novel views of a single generated subject are presented. It would be meaningful to evaluate and demonstrate multi-view consistency in this setting to further validate the quality and coherence of the reconstruction across viewpoints.

3) Animating Gaussian primitives using Linear Blend Skinning (LBS) may lead to noticeable artifacts, particularly for subjects wearing loose or complex garments such as dresses. This limits the realism and generalization of the animation results for diverse clothing types.

4) There appears to be a typographical error on line 82, which has an extra period at the end of the sentence. Kindly consider revising it for clarity.

---

> ### Author Rebuttal · Authors · 2025-07-31
>
> Thanks for your time and feedback. We also thank the reviewer for finding our method more applicable and useful for downstream tasks. We answer questions next.
>
> ***QE1. The paper does not provide qualitative results on in-the-wild images; all evaluations are conducted on datasets collected in controlled laboratory settings. Given that the method does not require pose information, it would be particularly valuable to demonstrate its performance on in-the-wild data to better assess its robustness and practical applicability.***
>
> Only plain text is allowed for the rebuttal, so we cannot show more rendering results. We instead refer to the presented results in our submission. In Part 4 of the supplementary webpage index.html, we demonstrate the cross-domain generalization on PeopleSnapshot and UBCFashion datasets. Among those datasets, UBCFashion consists of try-on videos like those on fashion retail websites, close to the in-the-wild scenarios. We kindly hope the reviewer reassesses the remarks on in-the-wild results.
>
> ***QE2. Only a limited number of novel views of a single generated subject are presented. It would be meaningful to evaluate and demonstrate multi-view consistency in this setting to further validate the quality and coherence of the reconstruction across viewpoints.***
>
> In Part 1 and Part 2 of the supplementary webpage index.html, we render 360 degree images and compare to baselines. The results demonstrate the multiview consistency as well as our improvements over baselines.
>
> ***QE3. Animating Gaussian primitives using Linear Blend Skinning (LBS) may lead to noticeable artifacts, particularly for subjects wearing loose or complex garments such as dresses. This limits the realism and generalization of the animation results for diverse clothing types.***
>
> LBS is a standard practice in pose-driven 3D animation: It is adopted by all the state-of-the-art baselines including LIFe-GoM, IDOL and LHM. LBS can handle most of the animations, see Part 3 in the supplementary webpage index.html. However, artifacts exist when retargeting loose garments to extreme poses, e.g., legs are widely open. We will revise the limitations to mention this point.
> We would like to emphasize that the main contribution of this paper is to eliminate the need for pose priors during reconstruction at test time. This is crucial since it is difficult to obtain accurate poses at test time, and pose-dependent methods are sensitive to the quality of the input poses. Meanwhile, the proposed method can also reconstruct loose clothes, as is shown in examples 4, 6, 7, 8 of Part 2 of the supplementary webpage.
>
> ***QE4. How does the model handle long garments, such as dresses, when predicting residuals in canonical pose? Are there specific mechanisms to account for the complex deformation of such clothing?***
>
> In the template branch, we predict residuals which accounts for the difference between the SMPL-X shape and the shape of an avatar with clothes, including complex garments. In the image branches, the model directly predicts the absolute positions that depict the shape of the dresses. We do not use any specific mechanism to model complex clothing. We find that this simple strategy works when trained on datasets that include complex clothing.
>
> ***QE5. Could the authors clarify the role of the canonical UV map in the pipeline? Specifically, how is it used, and what advantages does it provide?***
>
> The canonical UV map is the SMPL-X mesh in its average shape projected in the 2D UV space. It serves as the “initialization” of the Gaussians’ means in the template branch. Since the canonical UV map does not model the clothes and hair of the person, the template branch predicts the residuals of the means to the UV map to model those. Without the canonical UV map, the template branch is required to predict the absolute Gaussians’ means and LBS weights, which is more challenging than predicting the residuals.
>
> ***QE6. Does the method generalize well to animating subjects wearing dresses in diverse poses (especially for complex leg motion)? If so, could the authors provide examples or further discussion?***
>
> See QE3.
>
> ***QE7. The approach relies on a Gaussian primitive template to inpaint missing content in unseen regions, which could lead to blurry renderings when only a single image or multi-view images with limited coverage of the subject are available. Thus the method performs best when sufficient multi-view coverage is provided, while its effectiveness is reduced in sparse or narrowly-viewed scenarios.***
>
> Our approach can inpaint unobserved regions. We demonstrate this in Part 2 of the supplementary webpage index.html, where models only take a single front-facing image as input and the entire back side requires inpainting. In the first example, our approach hallucinates the wrinkles on the back while LHM fails. As mentioned in the limitations, we agree that the inpainting can be blurry. We think it is mainly due to the regression-based losses during the training. This is a shared issue in IDOL, LHM, LIFe-GoM, and our method. A possible solution is to use diffusion models to inpaint high-frequency details, which we leave for future work.

---

> ### Comment · Reviewer_Y9Hx · 2025-08-05
>
> Thank the author for clear explanation. Please try to include some qualitative result comparison to other baselines for in the wild images in the main paper as this will complete a more comprehensive evaluation, to help the reader understanding the performance of the proposed method.

---

### Official Review · Reviewer_7M22 · 2025-06-23

**Clarity:** 3
**Significance:** 3
**Originality:** 3
**Rating:** 5
**Confidence:** 4

**Summary:**

This paper proposes NoPo-Avatar, a new method for reconstructing animatable human avatars from sparse images without requiring any human or camera pose information. In contrast to most existing methods, NoPo-Avatar predicts a canonical T-pose avatar directly from images. The method contains two branches. The template branch reconstructs unseen regions by leveraging prior knowledge, while image branches focus on modeling visible details. Extensive experiments on multiple datasets show that the method outperforms existing methods like LHM.

**Questions:**

-  Unlike LHM, your method does not include a dedicated module (input) for facial modeling. Have you observed any degradation in quality or identity shift, specifically in the face region?

**Ethical Concerns:**

["NO or VERY MINOR ethics concerns only"]

**Final Justification:**

The authors' second rebuttal can address my concerns. I would like to keep the rating as "accept"

**Limitations:**

While most limitations are discussed in the supplementary materials, I believe there are additional concerns worth noting. One issue is identity preservation — the reconstructed avatars may exhibit identity shifts, which could hinder real-world applications. Moreover, accurately reconstructing and animating loose clothing remains an open challenge in this field and is not yet adequately addressed by the proposed method.

**Paper Formatting Concerns:**

I cannot see any major formatting issues in this paper.

**Quality:**

3

**Strengths And Weaknesses:**

**Strength**

- Novelty

  - The dual-branch architecture combining a learned template and image-aligned Gaussians is an interesting design and looks effective.

- Technical soundness

  - The architecture and training procedure are well-motivated and carefully designed.

  - Ablation studies convincingly demonstrate the necessity of each component.

- Evaluation results

  - Strong results across several datasets show broad generalization.

  - Comparisons with both pose-free methods (IDOL, LHM) and pose-dependent methods (LIFe-GoM, GHG) are extensive and fair.

- Discussion on Failure Cases

  - The paper has analyzed failure cases in the ablation study, providing further insights about the limitations of the work and the possibility of future work.

- Practical applications

  - The method is well-suited for real-world deployment in AR/VR scenarios due to its pose-free test-time pipeline and fast inference.

**Weaknesses**
-  The foundation of this paper is somewhat unclear. In the introduction, the authors primarily compare their method against previous approaches that rely on ground-truth or accurately estimated poses, claiming superior performance due to the removal of this dependency. However, recent methods such as LHM and IDOL also eliminate the need for pose inputs. Although the proposed method outperforms these baselines in the experiments, the authors do not provide sufficient analysis or explanation for this performance gain in the introduction.

- In the quantitative results, the method is not compared against IDOL and LHM on the THuman dataset.

---

> ### Author Rebuttal · Authors · 2025-07-31
>
> Thanks for your time and feedback. We also thank the reviewer for finding our method well-motivated, effective and interesting, and the results strong.
>
> ***QD1. The foundation of this paper is somewhat unclear. In the introduction, the authors primarily compare their method against previous approaches that rely on ground-truth or accurately estimated poses, claiming superior performance due to the removal of this dependency. However, recent methods such as LHM and IDOL also eliminate the need for pose inputs. Although the proposed method outperforms these baselines in the experiments, the authors do not provide sufficient analysis or explanation for this performance gain in the introduction.***
>
> Thanks for pointing this out! We did not mention LHM and IDOL in the introduction since both works focus on large-scale training of human models, not pose-free reconstruction. Instead, we opted to highlight them in the related work section. We will revise our paper and emphasize the differences in the introduction.
>
> ***QD2. In the quantitative results, the method is not compared against IDOL and LHM on the THuman dataset.***
>
> We follow the split of prior works (e.g., GHG and LIFe-GoM) and compare to multiview baselines on THuman. While IDOL provides the checkpoint in its official repository and the metrics on THuman + HuGe100K’s test set in the paper, we are not able to fairly compare to it since 1) our single-view model reported in Tab. 3 is trained on HuGe100K solely, but IDOL’s checkpoint is trained on HuGe100K + THuman; and 2) we are not able to finish the training on HuGe100K + THuman due to limited resources. Moreover, LHM does not provide the training script so we will only evaluate it on the same setting without training, using the same way as in Tab.3. We will add the comparisons in our final paper.
>
> ***QD3. Unlike LHM, your method does not include a dedicated module (input) for facial modeling. Have you observed any degradation in quality or identity shift, specifically in the face region?***
>
> We do not observe identity shift or degradation in face quality, even without a dedicated facial module, as can be seen in Part 2 of the supplementary webpage index.html. Note that our image branches output pixel-aligned Gaussians, which model more details than the SMPLX template branch, especially for fine-grained regions like faces.
>
> ***QD4. While most limitations are discussed in the supplementary materials, I believe there are additional concerns worth noting. One issue is identity preservation — the reconstructed avatars may exhibit identity shifts, which could hinder real-world applications. Moreover, accurately reconstructing and animating loose clothing remains an open challenge in this field and is not yet adequately addressed by the proposed method.***
>
> Please see QD3 for the identity shift. Our approach can reconstruct loose clothing reasonably, e.g., examples 4, 6, 7, 8 in Part 2 of the supplementary webpage index.html. Our approach adopts LBS for animation. Since loose clothes do not strictly follow LBS, sometimes it may fail to animate challenging poses. This is a common issue, also in baselines such as LIFe-GoM, IDOL and LHM, which we leave for future work. We’ll revise the limitation section to discuss this.

---

> ### Comment · Reviewer_7M22 · 2025-07-31
>
> The authors' rebuttals cannot convince me.
>
> 1. The authors say in the rebuttal: **Thanks for pointing this out! We did not mention LHM and IDOL in the introduction since both works focus on large-scale training of human models, not pose-free reconstruction.**
>
> However, this argument is unconvincing to me. While it is true that LHM and IDOL do not explicitly target "pose-free reconstruction," they have already demonstrated the capability to achieve it effectively. Therefore, a thorough analysis is essential—either in the introduction or the experimental section—to clearly explain why the proposed method outperforms LHM and IDOL. General or vague statements such as "we will revise the paper and emphasize the differences" are insufficient. The rebuttal must explicitly articulate the technical distinctions and provide evidence or reasoning (maybe without the need of solid experiments) as to why the proposed approach is superior comparing with LHM and IDOL.
>
> 2. The response to QD2 is unconvincing. In Table 3 of the paper, IDOL is successfully trained on HuGe100K for comparison—so why is it not possible to train IDOL on THuman as well? Moreover, the proposed method is trained on both THuman (Table 2) and HuGe100K (Table 3), yet the rebuttal claims that training on both datasets altogether introduces resource limitations. This contradiction needs to be addressed clearly. Furthermore, in Section 4 of the website provided by the authors, it is explicitly stated that "The model is trained on THuman2.1 + HuGe100K," which directly contradicts the claims made in the rebuttal.
>
> As for LHM, it should still be feasible to follow the same evaluation protocol as in Table 3 and report its performance on the THuman dataset. These comparisons are critical for a fair and thorough evaluation and should not be omitted without a concrete justification.
>
> 3. It is difficult to assess identity shift from the results in the supplementary material, as the faces are rendered too small. Close-up views are necessary for clearer visualization and meaningful comparison, particularly on real-world (THuman) and in-the-wild examples. At the very least, the THuman results (section 3 in webpage) show a noticeable drop in facial quality, and the cross-domain results (e.g., People-Snapshot in section 4 of webpage) show clear identity shifts—even the head shape changes significantly.

---

> > ### Author Response · Authors · 2025-08-02
> >
> > ***Distinctions and evidence why the approach is superior to LHM/IDOL.***
> >
> > Thanks for feedback and apologies for misunderstanding QD1 and for not being explicit. We’ll revise the introduction and add in L51 (also see L83-88 for the differences between our method and LHM/IDOL):
> > “Recent works such as LHM and IDOL scale training to improve generalization, and, like our method, eliminate the need for pose data. However, unlike IDOL and LHM which operate on a single input image, we support multiple input views. More importantly, we introduce a dual-branch design consisting of one template branch and one branch for images (which we refer to as “image branch”). The image branch predicts visible details because one Gaussian is used for each pixel in each image. The template branch is primarily used to inpaint unobserved regions. In contrast, LHM and IDOL rely solely on a template branch, which subtly differs from ours in the template encoding and the fusion of template embeddings and image features. We find the image branch enhances reconstruction of details in observed regions.”
> >
> > To quantify how the image branch enhances reconstruction we’ll add in L273 (experiments):
> > “Benefitting from the image branch, our method captures observed details better. To verify we assess results on HuGe100K data separately for all-observed views (front-facing) and for views with little/no overlap with the input image (three back views, camera rotated by at least 120 degree). PSNR/LPIPS*/FID results are:
> > | | All-observed views | Views with little/no overlap with the input image |
> > |--|--|--|
> > | LHM-1B | 18.56/105.86/23.25 | 17.17/138.34/37.27 |
> > | IDOL | 23.16/88.23/20.30 | 20.38/119.87/26.99 |
> > | Ours | 26.64/49.51/14.81 | 22.53/100.50/24.14 |
> >
> > The larger gain in all-observed views than views with little/no overlap with the input image (in LPIPS*, -38.72 vs -19.37 over IDOL, -56.35 vs -37.84 over LHM) highlights efficacy of our image branch in capturing observed details.”
> >
> > ***Why not train IDOL on THuman? Method is trained on THuman and HuGe100K, yet rebuttal says training on both has issues. Contradiction? In Sec. 4 of website: "The model is trained on THuman2.1+HuGe100K". Contradiction? For LHM, report results on THuman.***
> >
> > As suggested, we compare to LHM on THuman using the single image setting. Our method has PSNR/LPIPS*/FID of 24.64/49.69/34.82, LHM-1B has 22.03/70.31/63.34. Note, this comparison is not fair because LHM is not trained on THuman while ours is.
> >
> > To compare to IDOL we are pursuing two ways:
> >
> > 1\) Finetuning our method on THuman(IDOL’s split)+HuGe100K from our model in Tab.3. We’ll report results within the reviewer-author discussion period. This can be used as a preliminary comparison because IDOL provides an official checkpoint, trained on the same data. Note, finetuning rather than training from scratch may potentially harm results due to data imbalance. However, training from scratch on THuman(IDOL’s split)+HuGe100K jointly takes ~12 days and we can’t finish within the rebuttal+discussion period given limited access to 80GB GPUs.
> >
> > We cannot fairly compare the checkpoint used in Part 4 of the webpage to IDOL for two reasons: 1) Our model is trained and tested on 3 input images, while IDOL operates on one view. 2) There is train-test-set contamination because we adopt LIFe-GoM’s split on THuman, for a fair comparison in Tab. 6, while IDOL was trained on its own split.
> >
> > 2\) We are trying to compare with IDOL on THuman(LIFe-GoM’s split), following the single-view setting in Tab. 4. Training IDOL on THuman(LIFe-GoM’s split) is non-trivial: IDOL doesn’t provide preprocessing for THuman. Since IDOL only handles “neutral” SMPL-X, while THuman provides “male” SMPL-X, a conversion is needed. This involves solving an optimization, requiring ~3min for each of the 2445 subjects on a L40S. Still, due to the smaller data size we hope preprocessing+training will finish within the discussion period.
> >
> > ***Identity shift***
> >
> > We agree, close-up views permit to assess identity shift. We’ll revise the paper to add face zoom-in comparisons. To validate our claim, we assess identity shift quantitatively via the facial verification tool DeepFace. It verifies if two images show the same person. We think this is a good surrogate for identity shift. We adopt all 400 front-facing HuGe100K images and report the verified-rate, i.e., the percentage of “rendered image”-”GT image”-pairs recognized as the same person, and the cosine distance between the rendered face and the GT image in VGG-Face’s feature space. Our method without dedicated face module achieves a verified-rate of 95.25% and an average cosine distance of 0.32. LHM-1B has 95.75%/0.30. Note, LHM uses a dedicated face image crop as additional input and a feature pyramid for improved facial modeling. Both ours and LHM outperform IDOL’s 93.75%/0.45. We think this shows: our model does not suffer from a severe identity shift as results are comparable to methods with dedicated face module.

---

> > > ### Comment · Reviewer_7M22 · 2025-08-02
> > >
> > > Thank the authors very much about the detailed explaination. Please be sure to add the additional experimental results in the revised version.

---

> > > ### Author Response · Authors · 2025-08-09
> > >
> > > Thanks again for the suggestion on comparisons to IDOL on THuman. As explained in our previous response, these experiments take longer to complete. Here we provide the results we obtained.
> > >
> > > We conducted two types of comparisons to IDOL:
> > >
> > > 1\) Finetuning on THuman(IDOL’s split)+HuGe100K: We finetuned our method for 70K iterations, starting from the model whose results are reported in Tab. 3. On THuman (IDOL’s split), our finetuned method achieves PSNR/LPIPS*/FID = 21.39/122.13/90.05, improving upon IDOL’s 20.07/147.03/94.96. Note, joint training of our model on THuman(IDOL’s split)+HuGe100K may further improve results, but the full training process requires at least 12 days, which isn’t feasible to complete in the rebuttal and discussion period.
> > >
> > > 2\) Training IDOL on THuman(LIFe-GoM’s split): Following the single-view setting in Tab. 4 and using a white background following IDOL’s default, our method achieves a PSNR/LPIPS*/FID of 24.64/49.69/34.82, while IDOL obtains 23.47/66.62/83.37. Our approach shows better quantitative results in this setting.

---

### Official Review · Reviewer_LYXM · 2025-06-23

**Clarity:** 4
**Significance:** 3
**Originality:** 3
**Rating:** 5
**Confidence:** 3

**Summary:**

This paper proposes a method called NoPo-Avatar, which aims to reconstruct animatable 3D human avatars from single or sparse image sets without relying on human pose information. By eliminating the dependence on input poses, this method effectively addresses the issue of significant quality degradation in existing methods when pose estimation noise is present, enhancing the model's robustness and generalization ability in practical applications. NoPo-Avatar employs a dual-branch model, including a template branch and an image branch. The template branch captures the overall human structure and reconstructs unseen areas, while the image branch focuses on predicting Gaussian distributions aligned with the input images to capture details. Experiments show that this method achieves excellent performance on datasets such as THuman2.0, XHuman, and HuGe100K. It outperforms existing baselines in practical settings without ground-truth poses and matches methods using ground-truth poses in lab settings.

**Questions:**

1. Can you show the requirements for input images in real-world scenarios, such as whether the pose needs to be standing, and how the results are in such cases?
2. Are there any better suggestions for evaluating the accuracy of faces and hands?
3. During training, is the target pose for rendering GT, or is it robust to noise? Is there any data to support this?

**Ethical Concerns:**

["NO or VERY MINOR ethics concerns only"]

**Final Justification:**

Final Justification: After reading the rebuttal and author responses, I find the authors have addressed all my concerns regarding dataset generalization and robustness to pose noise. I maintain my score of 5 (Accept) as the paper remains technically sound and impactful.

**Limitations:**

Yes

**Paper Formatting Concerns:**

No formatting issues

**Quality:**

3

**Strengths And Weaknesses:**

Strengths
1. Quality: The paper is technically sound in its implementation. The proposed NoPo-Avatar method outperforms existing technologies in experimental results across multiple datasets, demonstrating its advantages in reconstruction quality and robustness. The authors provide detailed experimental settings and comparative analyses to clearly demonstrate the effectiveness of the method.
2. Clarity: The paper is well-organized and logically coherent. The entire story is narrated fluently, and the figures intuitively display the structure and characteristics of the method.
3. Significance: This research holds significant practical implications for fields such as virtual reality (VR) and augmented reality (AR).
4. Originality: NoPo-Avatar demonstrates innovation in several aspects: 1) Unlike previous methods, it does not require input image pose information, which is a significant advancement because such information is often inaccurate in real-world scenarios. 2) The dual-branch model design cleverly combines template priors and image details, ensuring the model's adaptability to sparse inputs while generating high-quality details.

Weaknesses
1. Dataset limitations: The datasets used for training and testing include THuman2.0, THuman2.1, and HuGe100K, with additional testing on XHuman. However, I would like to see more real-world, in-the-wild scenarios evaluated, as well as any limitations on the input person's actions, such as whether they can only be in a standing position, etc. These aspects are crucial for practical applications.
2. Insufficient detail modeling: In the results presented by the authors, the details of hands and faces are lacking. The authors also mentioned the reasons in the limitations section, such as the small part of image occupied by hands which casues challenges in predicting LBS weights and corresponding 3D locations in the canonical T-pose for each pixel in the image branches, and also possible solutions, train extra models for hands and faces beside the body part. Considering the importance of faces and hands in practical applications, I would like to ask if there are any appropriate metrics to evaluate this part for future research?
3. Rendering condition limitations: Although the input does not require poses, the rendering still needs GT poses. Is this GT pose the same size skeleton as the person being rendered? If so, this may not be the case in practical applications. Is it robust under noisy conditions?

---

> ### Author Rebuttal · Authors · 2025-07-31
>
> ***QC1. Dataset limitations: The datasets used for training and testing include THuman2.0, THuman2.1, and HuGe100K, with additional testing on XHuman. However, I would like to see more real-world, in-the-wild scenarios evaluated, as well as any limitations on the input person's actions, such as whether they can only be in a standing position, etc. These aspects are crucial for practical applications.***
>
> Only plain text is allowed for the rebuttal, so we cannot provide more rendering results. We instead refer to the presented results in our submission. In Part 4 of the supplementary webpage index.html, we demonstrate cross-domain generalization on PeopleSnapshot and UBCFashion datasets. Among the datasets, UBCFashion consists of try-on videos like those on fashion retail websites, close to in-the-wild scenarios.
> Our approach is not limited to standing poses. As results on THuman2.0 data show, our method performs well on input images showing diverse poses.
>
> ***QC2. Insufficient detail modeling: In the results presented by the authors, the details of hands and faces are lacking. The authors also mentioned the reasons in the limitations section, such as the small part of image occupied by hands which causes challenges in predicting LBS weights and corresponding 3D locations in the canonical T-pose for each pixel in the image branches, and also possible solutions, train extra models for hands and faces beside the body part. Considering the importance of faces and hands in practical applications, I would like to ask if there are any appropriate metrics to evaluate this part for future research?***
>
> Note that the details are lacking when hands and faces are heavily occluded. Otherwise, the image branches which predict pixel-aligned Gaussians depict details. To evaluate the hands and faces, a possible solution is to compute the PSNR/SSIM/LPIPS* in cropped regions based on hand and face masks.
>
> ***QC3. Rendering condition limitations: Although the input does not require poses, the rendering still needs GT poses. Is this GT pose the same size skeleton as the person being rendered? If so, this may not be the case in practical applications. Is it robust under noisy conditions?***
>
> During rendering, we require users to provide poses for animation. These can be any desired poses. We use the GT poses in the tables of the paper only for evaluation purposes, so that we can compare the renderings with the ground-truth images and baselines. This evaluation setting is identical to prior baselines.
>
> ***QC4. Can you show the requirements for input images in real-world scenarios, such as whether the pose needs to be standing, and how the results are in such cases?***
>
> See QC1.
>
> ***QC5. Are there any better suggestions for evaluating the accuracy of faces and hands?***
>
> A possible solution is to compute the PSNR/SSIM/LPIPS* in cropped regions based on hand and face masks.
>
> ***QC6. During training, is the target pose for rendering GT, or is it robust to noise? Is there any data to support this?***
>
> We use the GT poses provided by the dataset during training. To analyze the robustness to poses during training, we add synthetic Gaussian noise. We find that our approach is robust to some noise, e.g., Gaussian noise of std=0.1: On THuman2.0, it achieves a PSNR/LPIPS*/FID of 22.59/110.12/49.39, compared to 22.49/105.45/42.19 with GT training poses reported in Tab. 1. If we further perturb the GT poses with more significant Gaussian noise of std=0.3, the PSNR/LPIP*/FID drops to 20.58/138.67/73.20.
> We do not think robustness to the target poses in training is a big concern. Collecting high-quality training data is a one-time effort. Once trained, the model does not rely on poses for reconstruction at inference time. This is different from pose-dependent methods, which still require high-quality input poses during inference for reconstruction.

---

> > ### Comment · Reviewer_LYXM · 2025-08-05
> >
> > I think the authors well addressed the main concerns.
> > 1. For QC3 (robustness to pose noise during training), they provided clear quantitative results showing that the model still performs well even when Gaussian noise is added to the training poses. That really helped clarify things and gave me confidence in the method’s robustness.
> > 2. Regarding QC1 and QC4, the authors showed that the method isn’t limited to standing poses and generalizes to more realistic, in-the-wild scenarios like PeopleSnapshot and UBCFashion. That makes it more practical and applicable in real-world settings.
> > 3. For QC2 and QC5 (lack of facial and hand detail), the authors acknowledged this limitation and also suggested reasonable future directions, like evaluating on cropped hand/face regions or even training separate modules for those parts.
> > Overall, I’m satisfied with the response — the explanations were clear, and the experiments backed them up well. So I keep my original score (Accept).

---

### Official Review · Reviewer_VStF · 2025-07-01

**Clarity:** 2
**Significance:** 3
**Originality:** 2
**Rating:** 4
**Confidence:** 4

**Summary:**

This method does not require camera pose or human pose information corresponding to the input image during the reconstruction phase, and can reconstruct drivable 3D human avatars from sparse image inputs. Paper claims it addresses the critical issue of degraded reconstruction quality in existing methods due to their reliance on inaccurate pose estimation.

**Questions:**

Reconstruction stability regarding standard poses. Results for complex input images (even for use cases where pose estimators trained specifically for them have problems).

Comparison with methods using high-precision pose input in the case of simple input images.

Provide scaling experiments regarding computational cost.

Current results are good with sparse image input, but overall there is a lot of room for improvement. With better input conditions, such as increasing the number of image inputs, what are the performance boundaries? Can it surpass existing methods?

Why use T-pose but not da-pose?

**Ethical Concerns:**

["NO or VERY MINOR ethics concerns only"]

**Final Justification:**

After reviewing the authors’ rebuttal and considering the reviews and responses from other reviewers, I have decided to raise my final rating.

**Limitations:**

Yes

**Quality:**

3

**Strengths And Weaknesses:**

Strengths:
1. The problem addressed by the paper is important, namely the reliance on accurate pose during the reconstruction stage. By resolving this dependency, the method enhances practicality and robustness in real-world scenarios.
2. The two-branch architecture is reasonable.
3. The paper conducts experiments on multiple datasets, which is good. It also demonstrates the superiority of the method.
test-time pose optimization is interesting.

Weaknesses:

Although the paper does not require pose, it does require providing a mask. This demands on the quality of the mask.

Applying a separate branch to each image may lead to an increase in computational cost.

There is a lack of complex action image input. The paper uses an internal normalization strategy instead of relying on GT pose. This is good, but it raises questions about the normalization process because it cannot guarantee overall performance compared to precise input pose or the normalization results under complex action image input.

---

> ### Author Rebuttal · Authors · 2025-07-31
>
> Thanks for your time and feedback. We also thank the reviewer for finding our method superior and the test-time pose optimization interesting. We answer questions next.
>
> ***QB1. Although the paper does not require pose, it does require providing a mask. This demands on the quality of the mask.***
>
> Mask prediction is usually more robust than pose estimation. This is intuitive because masks are directly related to input images while poses can be ambiguous, e.g., because of clothing or occlusion. Therefore, we aim to avoid poses as input.
> Moreover, our approach is robust to the mask quality. Note that we use the predicted masks for training and inference on HuGe100K. To further evaluate the robustness, we additionally corrupt the masks by dilating with kernels of size 3 and 5. Our approach achieves a PSNR/LPIPS*/FID of 22.92/93.04/15.69 when the dilation kernel size is 3, and 22.83/95.03/15.80 when the kernel size is 5. Compared to 23.15/90.63/15.56 reported in Tab.3, the FID scores stay almost the same, demonstrating the robustness of our approach w.r.t. mask quality.
>
> ***QB2. Applying a separate branch to each image may lead to an increase in computational cost.***
>
> Providing more input images increases computation but also improves quality, as is shown in Tab. 4. Note that our approach is significantly faster than LHM and comparable to IDOL, as shown in Tab. 3. Even with 3 input images of resolution 1024x1024, our approach still achieves 1.32s (Line 259), faster than LHM-500M’s 2.69s obtained with a single image of 896×640 resolution.
>
> ***QB3. There is a lack of complex action image input. The paper uses an internal normalization strategy instead of relying on GT pose. This is good, but it raises questions about the normalization process because it cannot guarantee overall performance compared to precise input pose or the normalization results under complex action image input.***
>
> We clarify that our method is evaluated on complex action image inputs using the THuman2.0 dataset, which includes challenging poses and occlusions. Pose estimators often fail on THuman2.0, degrading pose-dependent baselines—as shown in Part 1 of our supplementary webpage. This highlights the difficulty of relying on pose estimators for precise input pose under complex motion.
> Despite not using ground-truth poses, our method achieves comparable or better performance. In Tab. 2, we compare our approach to the baselines with precise GT poses and show comparable results with better LPIPS* and FID, indicating better visual quality. Therefore, the overall performance is compelling even for complex action image input. We would like to emphasize that in the real world, input poses are either less precise, or are time-consuming (e.g., 20min) to acquire. So it is impractical to assume precise input poses.
>
> ***QB4. Reconstruction stability regarding standard poses. Results for complex input images (even for use cases where pose estimators trained specifically for them have problems).***
>
> In Fig. 1 and in Tab. 7 of the appendix, we evaluate stability of approaches w.r.t. pose quality. We examine the stability by adding Gaussian noise with different standard deviations or by using predicted poses. Our approach is comparable to the baselines with _ground-truth_ poses as inputs. Please see QB3 for results on complex input images.
>
> ***QB5. Comparison with methods using high-precision pose input in the case of simple input images.***
>
> For simple input images where pose estimation is less likely to fail, our approach achieves superior visual quality compared to pose-dependent approaches. We compare our approach to LIFe-GoM on a subset of HuGe100K’s test data which contains 400 subjects in the simplest A-pose. Both methods are trained on THuman2.1+HuGe100K. Our approach achieves a PSNR/LPIPS*/FID of 22.69/98.80/20.65, significantly outperforming LIFe-GoM’s 18.28/121.63/26.12. We find that LIFe-GoM predicts spatially inconsistent colors due to view-dependent effects in the input images, an issue our method avoids. Additionally, we find LIFe-GoM performs worse than ours in inpainting unobserved regions.
>
> ***QB6. Provide scaling experiments regarding computational cost.***
>
> 1\) Scaling the training data as in Tab. 6 in the appendix won’t increase the computational cost during inference. 2) Increasing the number of input images from 1 to 3 increases reconstruction time from 495ms to 1.32s on one NVIDIA A100 when using a resolution of 1024x1024. But it also improves quality, e.g., LPIPS* from 124.36 to 106.98, as shown in Tab. 4.
>
> ***QB7. Current results are good with sparse image input, but overall there is a lot of room for improvement. With better input conditions, such as increasing the number of image inputs, what are the performance boundaries? Can it surpass existing methods?***
>
> Our paper focuses on the setting of sparse input images as it is more convenient for real-world applications. In this setting, we improve upon state-of-the-art methods. Our approach also demonstrates better performance with 3 input images compared to using a single image (Tab. 4). We expect further quality improvements with more input images since there is less to hallucinate. Our approach can operate on as many as 18 input images while using ~97GB GPU memory.  Unfortunately, due to the limited resources, we are unable to finish this training.
>
> ***QB8. Why use T-pose but not da-pose?***
>
> We use T-pose since it is the SMPL-X defined canonical space and LIFe-GoM also adopts it. Due to the limited rebuttal time, we are not able to finish the ablation. We suspect a different format won’t make a big difference, as the poses are related via a transformation that can be learnt by our deep nets.

---

### Official Review · Reviewer_KMYG · 2025-07-03

**Clarity:** 2
**Significance:** 3
**Originality:** 3
**Rating:** 4
**Confidence:** 5

**Summary:**

This paper introduces NoPo-Avatar, a novel method for reconstructing animatable 3D human avatars from sparse images without relying on human or camera pose priors. The proposed framework reconstructs avatars in canonical T-pose space using a dual-branch architecture: a template branch that captures global body structure and facilitates inpainting of unseen regions, and image branches that predict pixel-aligned Gaussians for visible details.

**Questions:**

See the weaknesses.

**Ethical Concerns:**

["NO or VERY MINOR ethics concerns only"]

**Final Justification:**

The author addressed my main concern, so I will keep my original score.

**Limitations:**

See the weaknesses.

**Quality:**

3

**Strengths And Weaknesses:**

# Strengths
I think the paper is overall well-executed — the results are solid, and the framework is intuitive and easy to understand.
1. NoPo-Avatar eliminates the need for test-time pose and camera pose input, increasing robustness and applicability. The design choice is well-justified through both qualitative and quantitative results.

2. The proposed template + image branch design is elegant and effective. Ablation studies confirm that the combination enhances both detail preservation and inpainting of unseen regions.
Extensive Evaluation

3. The paper includes comprehensive experiments against strong baselines (LIFe-GoM, IDOL, LHM, GHG, etc.) across multiple datasets and settings (with ground truth poses, predicted poses, no poses at all).

# Weaknesses

1. I would like to see more results on long dresses or loose clothing.
These cases are often challenging for avatar reconstruction, and it would help demonstrate the robustness of your method.

2. For the comparison with LHM and IDOL in Fig. 4, could you clarify what the input image is for each method?
Were the same images used across all methods? Additional detail here would improve the fairness and interpretability of the comparison.

3. A recent paper has tackled a similar problem. I would appreciate it if you could include a comparison with their results.
[1] (PF-LHM) It would help position your contributions more clearly within the current landscape of pose-free avatar reconstruction.

PF-LHM: 3D Animatable Avatar Reconstruction from Pose-free Articulated Human Images

---

> ### Author Rebuttal · Authors · 2025-07-31
>
> Thanks for your time and feedback. We also thank the reviewer for finding our method well-justified, elegant and effective, and the results solid. We answer questions below.
>
> ***QA1. I would like to see more results on long dresses or loose clothing. These cases are often challenging for avatar reconstruction, and it would help demonstrate the robustness of your method.***
>
> We find our method reconstructs long dresses or loose clothing well particularly if samples with loose clothes are part of the training data. We showcase an example of long dresses in Fig. 4. We also provided freeview rendering of long dresses in Part 2 of the supplementary file index.html.
>
> ***QA2. For the comparison with LHM and IDOL in Fig. 4, could you clarify what the input image is for each method? Were the same images used across all methods? Additional detail here would improve the fairness and interpretability of the comparison.***
>
> The input image is the front-facing image of the subject. The same image is used for all methods and the comparison is completely fair. We will emphasize this by revising Fig. 4.
>
> ***QA3. A recent paper has tackled a similar problem. I would appreciate it if you could include a comparison with their results. [1] (PF-LHM) It would help position your contributions more clearly within the current landscape of pose-free avatar reconstruction.***
>
> Thanks for the reference. We are happy to cite it in the final version. Please note that the paper appeared on arXiv on June 16, i.e., after the NeurIPs submission deadline. The codes have not been released yet. We compare our approach to its previous version (LHM) in Tab. 3.

---

> > ### Comment · Reviewer_KMYG · 2025-08-05
> > **Official Comment of Reviewer KMYG**
> >
> > The author addressed my main concern, so I will keep my original score.

---

### Comment · Area_Chair_iP6B · 2025-08-05

Dear Reviewers,

Thank you for your efforts. Please revisit the submission and check whether your earlier concerns have been adequately addressed in the author's response. If you haven't, please join the discussion actively before August 8 to let the authors know if they have resolved your (rebuttal) questions.

Best, AC

---

### Note · Authors · 2025-08-15

We propose NoPo-Avatar, a method for recovering an animatable 3D human avatar from a single or a sparse set of images __without any pose priors__. NoPo-Avatar outperforms existing pose-based baselines in practical settings (without ground-truth poses), and, in controlled lab settings it achieves comparable results to baselines that use ground-truth poses. It also surpasses recent pose-free reconstruction and rendering methods.

We thank reviewers for finding our method well-motivated, effective and applicable (Reviewer KMYG, VStF, LVXM, 7M22 and Y9Hx), results solid (Reviewer KMYG), and useful for downstream tasks (Reviewer Y9Hx and VStF).

In the rebuttal, we mainly clarified paper details and answered the following questions:
* Our method takes images showing a human in various poses as input. It is not limited to standing poses, as shown in our THuman2.0 results (QB3, QC1).
* Our method is robust to the accuracy of subject masks (QB1).
* Our method, without any pose priors, outperforms pose-based methods irrespective of whether they use ground-truth poses or simple poses (QB4, QB5).
* Our method can generalize to in-the-wild settings, as demonstrated by examples from the UBCFashion data, which consists of try-on videos like those on fashion retail websites, close to in-the-wild scenarios (QC1, QE1).
* Our method is able to reconstruct loose clothes (QA1, QE4). Following prior works (LIFe-GoM, IDOL, LHM, etc), we animate loose clothes with linear blend skinning. This can lead to artifacts (QD4, QE3). Like prior art, we leave it for future work.
* Our method outperforms pose-free baselines IDOL and LHM on THuman (QD2 and its followup discussion).
* Our method is robust to small noise in human poses during training (QC6).
* Our method, benefitting from the image branch, does not suffer from severe identity shift (QD3, QD4). Even with the image branch, the reconstruction speed of our method is faster than LHM and is comparable to IDOL (QB2, QB6).

We hope our responses address the reviewers’ questions. All additional results presented in the discussion will be included in the final version of our paper. Thanks a lot for your valuable feedback.

---

### Decision · Program_Chairs · 2025-09-17

**Decision:**

Accept (poster)

**Comment:**

This paper eliminates pose dependencies for avatar reconstruction while maintaining competitive performance. Reviewers agreed that the idea is novel and practical, the technical design is elegant, the evaluation is thorough and convincing, the writing is clear, and the applications are promising.

Initial concerns focused on comparisons with other **pose-free** methods, handling of challenging cases such as loose clothing, evidence on **in-the-wild** datasets, potential identity shift problem, and robustness consideration. Most of these issues were addressed during the rebuttal and discussion through the presentation of additional quantitative results and clarification. While some limitations remain (e.g., loose clothing, hand/face detail, and in-the-wild evaluation), the authors have acknowledged these issues and proposed potential solutions, which the reviewers found acceptable.

The final ratings converged to a positive consensus (3 borderline acceptances and 2 acceptances). Therefore, **it's a clear acceptance**, considering its contributions, quality of the rebuttal, and the consensus. The additional clarifications and comparisons from the discussion are requested to be incorporated into the final version.